# Beyond FVD: An Enhanced Metric for Evaluating Video Generation Distribution Quality

**Ge Ya Luo**[*]**, Gian Mario Favero**[*]**, Zhi Hao Luo**
Mila - Quebec Artificial Intelligence Institute

**Alexia Jolicoeur-Martineau**
Samsung - SAIT AI Lab, Montreal

**Christopher Pal**
Mila - Quebec Artificial Intelligence Institute
Canada CIFAR AI Chair

## Abstract

The Fréchet Video Distance (FVD) is a widely adopted metric for evaluating video generation distribution quality. However, its effectiveness relies on critical assumptions. Our analysis reveals three significant limitations: (1) the non-Gaussianity of the Inflated 3D ConvNet (I3D) feature space; (2) the insensitivity of I3D features to temporal distortions; (3) the impractical sample sizes required for reliable estimation. These findings undermine FVD's reliability and show that FVD falls short as a standalone metric for video generation evaluation. After extensive analysis of a wide range of metrics and backbone architectures, we propose **JEDi**, the **J**EPA **E**mbedding **Di**stance, based on features derived from a Joint Embedding Predictive Architecture, measured using Maximum Mean Discrepancy with polynomial kernel. Our experiments on multiple open-source datasets show clear evidence that it is a superior alternative to the widely used FVD metric, requiring only 16% of the samples to reach its steady value, while increasing alignment with human evaluation by 34%, on average.
**Project page:** `https://oooolga.github.io/JEDi.github.io/`.

## 1 Introduction

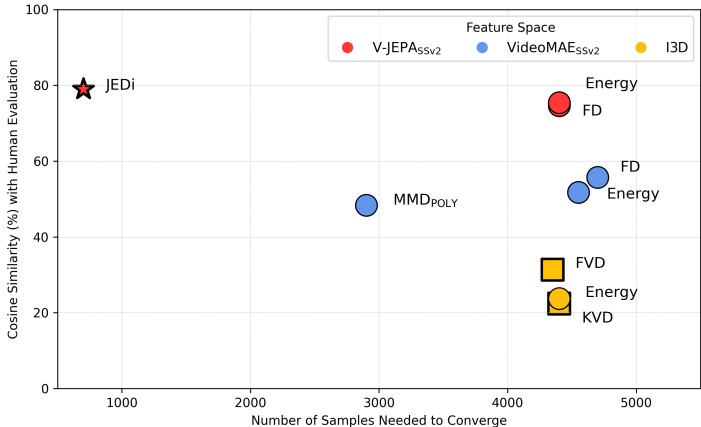

Figure 1: Comparing the number of samples that Fréchet Distance (FD), Energy, and $MMD_{POLY}$ need to converge against its alignment with human evaluation on the UCF-101 dataset. JEDi, the feature space of a V-JEPA model ($\text{V-JEPA}_{SSv2}$) in combination with a Maximum Mean Discrepancy (MMD) metric, is a vastly more efficient framework for evaluating distributions of generated videos than conventional methods. The current standard, FVD (FD+I3D), underperforms in terms of both sample efficiency and alignment with human evaluation.

Video generation research has experienced a significant surge recently, yielding cutting-edge models that produce high-quality videos (Liu et al., 2024; Blattmann et al., 2023; Zeng et al., 2023; He et al., 2023). However, evaluating their generation quality poses a substantial challenge.

A good video generation model must produce high-quality frames, maintain temporal consistency, and generate varied content. For instance, at the object level, it is undesirable to have exclusively cars of specific brands or colors when generating automobile videos; at the motion level, it is undesirable to observe the same type of motion repeatedly when generating human action videos. Therefore, a robust evaluation metric must address multiple facets to reliably assess generative video models.

Researchers have created a range of evaluation metrics and tools to assess the quality of individual outputs from video generation models. Many of the older video generation metrics are derived from image quality assessments: LPIPS, MSE, SSIM, PSNR. These metrics fail to quantify the temporal consistency between frames (Zhang et al., 2018; Horé & Ziou, 2010). Recent video distance metrics focus on assessing both temporal and spatial qualities, as well as diversity, by measuring distances in distribution with respect to the real videos. The most popular such metric for video is the Fréchet Video Distance (FVD) (Unterthiner et al., 2019)[1]. An analog exists for measuring image quality (but not temporal consistency): the Fréchet Inception Distance (FID) (Heusel et al., 2017). These metrics have emerged as a leading tool for assessing the quality of video and image generation models.

FID, primarily a image metric, is also used in video generation to compare key-frames. It computes the Fréchet distances of the frame features from Inception v3 (Szegedy et al., 2014; Ioffe & Szegedy, 2015) trained on ImageNet. Building upon the principles of FID metrics, FVD evaluates the FD between the generated and data distributions in the Inflated 3D ConvNet (I3D)'s feature space (Carreira & Zisserman, 2018), which is trained on the Kinetics dataset (Kay et al., 2017). FVD's use of features extracted from 3D-ConvNet allows it to capture a more comprehensive range of visual and temporal information compared to FID.

Recent studies have highlighted limitations in the reliability of the FVD measure. Specifically, Brooks et al. (2022) demonstrated that FVD is not effective in capturing long-term realism and is more suitable for comparing generation model variants of the same architecture. Moreover, Skorokhodov et al. (2021) showed that FVD overlooks motion collapse and is biased towards image quality, rather than video quality. Additionally, they pointed out that FVD is excessively sensitive to minor implementation details, such as the specific image storage formats used (e.g., JPEG compression levels or file encoding), which can lead to inconsistent and non-comparable results across different studies.

A comprehensive study on FVD was conducted by Ge et al. (2024), which compares Fréchet distances of features extracted by the I3D network (Carreira & Zisserman, 2018) and VideoMAE network (Wang et al., 2023a). The study shows that the FVD prioritizes per-frame quality over temporal consistency when using I3D features, which they refer to as content-bias. Further, they suggest that using features from self-supervised models trained on content-debiased data can effectively mitigate this bias in FVD. Our methodology draws inspiration from previous analysis that highlights shortcomings with FID (Borji, 2021; Kynkäänniemi et al., 2023; Soloveitchik et al., 2022; Sajjadi et al., 2018).

A separate and distinct method of evaluating videos is on the sample level, rather than the distributional level. For example, (Huang et al., 2023) recently developed VBench, a comprehensive video benchmark that analyzes the evaluation of individual generated outputs on subject consistency, background consistency, temporal flickering, motion smoothness, aesthetic quality, among others. VBench addresses the challenge of evaluating both temporal and spatial consistency in video generation, but naturally fails at efficiently evaluating the generational capabilities of a model on a distributional level, or the robustness of a model in generating videos outside its benchmark distribution.

In our proposed framework, **JEDi**, we address many of the problems affecting existing evaluation strategies:
1. JEDi employs a Maximum Mean Discrepancy (MMD) metric with a polynomial kernel, eliminating the need for parametric assumptions about the underlying video distribution, unlike FVD which relies on the Gaussianity assumption to make its metric feasible.

---

[1] Unterthiner et al.'s work introduced Fréchet Video Distance (FVD) and Kernel Video Distance (KVD), both operating in the I3D feature space. FVD uses the Fréchet distance, while KVD employs a polynomial kernel-based method, and Unterthiner et al. found that FVD aligns more closely with human judgment.

2. JEDi significantly reduces the number of samples needed to make an accurate estimate by using an MMD metric in a V-JEPA feature space, enabling reliable use in smaller datasets that do not meet the requirement when using FVD.

3. JEDi leverages the robust representations of a V-JEPA model, which are found to be more aligned with human evaluations compared to FVD.

## 2 BACKGROUND AND NOTATIONS

### 2.1 VIDEO FEATURE REPRESENTATION

**Inflated 3D ConvNet:** The Inflated 3D ConvNet (I3D) (Carreira & Zisserman, 2018) is a convolutional neural network model based on the pre-trained Inception-v1. It extends the 2D convolutional filters to 3D by replicating them along the temporal dimension. I3D, pre-trained on Kinetics, has demonstrated excellent classification performance on UCF-101 (Soomro et al., 2012), HMDB-51 (Kuehne et al., 2011), and Kinetics datasets (Kay et al., 2017), proving to be a valuable network for video recognition tasks.

The original FVD work by Unterthiner et al. (2019) explores the use of I3D features trained on the Kinetics datasets. They analyze the features from the logits layer, as well as the features from the last pooling layer trained on the Kinetics-400 and Kinetics-600 datasets. Their experiments suggest that the features from the logits layer trained on the Kinetics-400 dataset are the most suitable for the FVD metric.

**Video Masked Autoencoder:** The Video Masked Autoencoder (VideoMAE-v2) (Wang et al., 2023a) is a self-supervised pre-training method that leverages a vision transformer (ViT) backbone (Dosovitskiy et al., 2020) to learn efficient video representations. According to Ge et al., the giant-VideoMAE-v2 model, pretrained on a diverse set of unlabeled datasets and fine-tuned on Something-something-v2 (Goyal et al., 2017) with a masked autoencoder objective, effectively captures both spatial and temporal distortions in its encoded feature space. We leverage two variants of the VideoMAE-v2 model in our study: (1) *VideoMAE$_{PT}$*: the self-supervised pre-trained giant VideoMAE-v2 model and (2) *VideoMAE$_{SSv2}$*: the fine-tuned giant VideoMAE-v2 model.

**Video Joint Embedding Predictive Architecture:** Video Joint Embedding Predictive Architecture (V-JEPA) (Bardes et al., 2024) is a self-supervised training paradigm that learns by predicting missing or masked parts of a video in an abstract representation space. V-JEPA excels in "frozen evaluations", where its encoder and predictor are pre-trained through self-supervised learning and then left unchanged. For new tasks, only a small, lightweight layer or network is trained on top of the pre-trained components, enabling quick and efficient adaptation to new environments. In this study, we employ both (1) the pre-trained variant of the model trained with the self-supervised objective, *V-JEPA$_{PT}$*, as well as (2) a version that was fine-tuned on Something-something-v2 (Goyal et al., 2017) with an attentive classification probe such that its pre-logit features could be used for distributional analysis metrics, *V-JEPA$_{SSv2}$*.

### 2.2 FRÉCHET DISTANCE AND FRÉCHET VIDEO DISTANCE

Fréchet Distance (FD), also known as 2-Wasserstein distance ($W_2$), is a way of measuring how similar two distributions are (Frechet, 1957; Dowson & Landau, 1982; Zilly et al., 2020). The Fréchet distance between two distributions $P$ and $Q$ is defined as the minimum distance between all pairs of random variables x and y from the distributions. Assuming $P$ and $Q$ are multivariate Gaussian distributions, it can be expressed as:

$$D^2_{\text{Fréchet}}(P, Q) = (\mu_P - \mu_Q)^2 + \text{Tr}(\Sigma_P + \Sigma_Q - 2(\Sigma_P \Sigma_Q)^{\frac{1}{2}}) \qquad (1)$$

where $\mu_P$ and $\mu_Q$ are the means, while $\Sigma_P$ and $\Sigma_Q$ are the covariance matrices of the two Gaussian distributions. Without making this assumption, the Fréchet Distance is intractable and becomes much more arduous to obtain.

The Fréchet Inception Distance (FID) and Fréchet Video Distance (FVD) correspond to the above equations, but the distance is applied in the space of InceptionV3 and I3D network features, respectively, instead of directly in raw image space in order to obtain more meanful distance that better align with human preferences (Szegedy et al., 2014; Unterthiner et al., 2019).

## 2.3 OTHER DISTRIBUTION DISTANCE METRICS

This study also explores the application of alternative statistical methods to compute probability distribution distances in video feature spaces, including Mixture Wasserstein ($MW_2$), Energy Statistics, and kernel-based methods such as Maximum Mean Discrepancy (MMD). The detailed backgrounds of these metrics are provided in Appendix A.4.

## 3 EXAMINING FVD: FEATURE SPACES AND THE GAUSSIANITY ASSUMPTION

The Fréchet distance (FD) measures the difference between means and covariances. This can offer insights into the first two moments of the distributions, but fails to do so with respect to higher-order moments (e.g., skewness, kurtosis) that arise when either the real or generated data distribution is non-Gaussian. According to Jayasumana et al. (2024), the reliance on Gaussianity assumptions in FID research can lead to substantial inaccuracies when the underlying image distribution does not come from such a distribution. This part of the study focuses on the video feature spaces, investigating the accuracy of Gaussian assumptions and considering the consequences of the Fréchet Video Distance (FVD) when those assumptions are not met.

Using each of the I3D, VideoMAE, and V-JEPA networks under comparison, we extract 48,501 features from 11 distinct video datasets, with each feature representing a 32-frame clip. Specifically, we extract a maximum of 5000 features from the training set of each dataset, which include: Anime-Run-v2 (Siyao et al., 2022), BAIR (Ebert et al., 2017), BDD100k (Yu et al., 2020), DAVIS (Pont-Tuset et al., 2018), Fashion Modeling (Zablotskaia et al., 2019), HMDB-51 (Kuehne et al., 2011), How2Sign (Duarte et al., 2021), KITTI (Geiger et al., 2013), Something-Something-v2 (Goyal et al., 2017), Sky Scene (Xiong et al., 2018), and UCF-101 (Soomro et al., 2012).

The notion that I3D video features do not follow multivariate Gaussian distributions is investigated using the widely-accepted Mardia's Skewness (Mardia, 1970), Mardia's Kurtosis (Mardia, 1970), and Henze-Zirkler normality tests (Henze & Zirkler, 1990), following (Jayasumana et al., 2024). The null hypothesis that I3D features follow a multivariate Gaussian distribution is strongly rejected ($p = 0$) across all datasets and normality tests.

We then normalize the aggregated training-set features and fit a Principal Component Analysis (PCA) model and a Linear Discriminant Analysis (LDA) model using dataset labels as classes. We apply the same pipelines to transform 5,256 I3D features (up to 500 samples from each of the eleven datasets' *testing sets*) into lower-dimensional spaces for visualizations. As demonstrated in Appendix A.3, applying PCA or LDA transformations to Gaussian-distributed data preserves their Gaussian properties. Figure 2 shows that the I3D features don't follow a single multivariate Gaussian distribution; rather, they cluster by dataset.

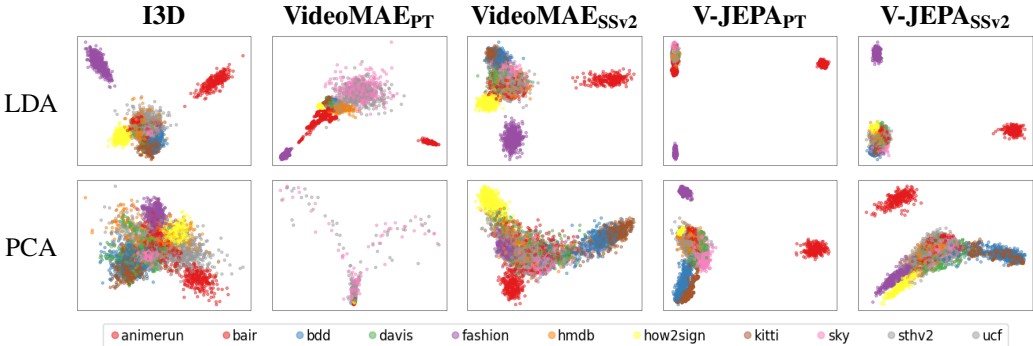

Figure 2: The dimensionally reduced video features of the 11 datasets using LDA and PCA indicate that the video features are non-Gaussian in the combined dataset space. While individual dataset clusters may appear Gaussian in these plots, the low explained variance ratios (0.134-0.231) of the PCA-reduced spaces suggest that 2D projections in these plots may not capture the complexity of higher-dimensional feature distributions within individual datasets. Figures 7 and 8 contain dataset-specific LDA and PCA plots, which reveal non-Gaussian characteristics within the datasets.

Moreover, we conduct a PCA on individual datasets and apply a LDA to the HMDB-51 and UCF-101 datasets, incorporating classification labels and varying frame counts per clip. Our analysis shows that I3D clip features deviate from Gaussian distributions within each dataset. More notably, we observe a *positive correlation between clip duration and increase in FVD between the train and test sets from the same dataset*, suggesting that higher-order moments may be essential for accurate characterization.

We replicate our experiments in the VideoMAE and V-JEPA feature spaces using the same datasets. Our results in these feature spaces mirror our findings in the I3D feature space. Additional details on the setup and results are provided in Appendix D.

## 4 THE DUAL CHALLENGE OF CONVERGENCE: HIGH-DIMENSIONAL FEATURE SPACES AND LIMITED SAMPLES

The following section addresses two pivotal challenges in evaluating video distribution distances:

1. *The Dimensionality Problem* (Section 4.1): We examine the limitations of metrics relying on distribution assumptions (e.g., Fréchet distance, Mixture Wasserstein distance), highlighting the adverse impact of high dimensionality.

2. *Sample Efficiency and Convergence* (Section 4.2): We discuss the sample efficiency issue affecting all metrics and the necessary sample size for trustworthy measurements.

### 4.1 CHALLENGE #1: THE CURSE OF DIMENSIONALITY

#### IMPACT OF DATA DIMENSION ON FRÉCHET DISTANCE METRIC

In the previous section, we have shown that the Fréchet Distance (FD) can be used as a metric for comparing the discrepancy between the first two moments of two distributions. Consequently, the accuracy of the mean and covariance estimators is crucial for ensuring the validity of FD as a metric. In the following part, we will explore the impact of data dimensionality and sample size on the quality and precision of these estimators, and examine how this affects the reliability of distribution distance metrics.

The rank of the empirical covariance matrix ($\hat{\Sigma}$) is tied to the number of samples ($n$) and the dimension ($k$). Given a matrix $\mathbf{X}$ containing $n$ observations, where each column vector represents a $k$-dimensional multivariate sample, the empirical covariance matrix can serve as a reliable estimator for the true covariance matrix. The empirical covariance matrix is calculated using the formula: $\hat{\Sigma} = \frac{1}{n} \sum_{i=1}^{n} (\mathbf{X}_i - \bar{\mathbf{X}})(\mathbf{X}_i - \bar{\mathbf{X}})^{\mathsf{T}}$, and it consistently converges to the true $\Sigma$ at a rate of $\frac{1}{\sqrt{n}}$.

It is crucial to recognize that when the number of samples is less than the number of variables ($n < k$), the covariance matrix becomes singular. A good covariance estimator requires a sample size that is sufficiently large, ideally at least several times greater than the data dimension. This is because estimating the covariance matrix involves estimating $k(k+1)/2$ parameters, which requires a sufficiently large number of samples to achieve accurate estimates (Bickel & Levina, 2008; Marčenko & Pastur, 1967; Wang; Jonsson, 1982). Unfortunately, the high-dimensional nature of I3D (400), VideoMAE (1408), and V-JEPA (1280) representation spaces exacerbates this issue.

Furthermore, optimal transport methods with complex distributional assumptions require more samples yet. The Mixture Wasserstein ($MW_2$) experiment described in Appendix E.1 highlights significant computational and practical limitations of optimal-transport type metrics, making them impractical for this project. See Appendix E.1 for further discussion.

#### DATA TRANSFORMATION: DIMENSIONALITY REDUCTION

To address the challenges posed by the curse of dimensionality, dimension reduction techniques such as PCA and autoencoders (Lecun, 1987) can be applied. Our preliminary investigation from Appendix E.1 suggests that decreasing the representation dimension could enable the metric to converge with a smaller number of samples using metrics like Fréchet Distance. We test that hypothesis by training autoencoders in various feature spaces to reduce dimensionality.

Our autoencoder architectures consist of simple multilayer perceptron networks, which compress feature dimensionality to either $\frac{1}{6}$ of original size for I3D features or $\frac{1}{8}$ of original size for Video-MAE and V-JEPA features. Additional information about the autoencoder training is available in Appendix E.2.

Interestingly, dimension reduction significantly enhances the sample efficiency of the Fréchet Distance and energy statistic metrics in our experiments with Gaussian data (Figure 10). However, its benefits are less pronounced for video features, resulting in only marginal improvements (Figures 11 and 12). Nonetheless, we retained autoencoder features in subsequent experiments to investigate other possible benefits.

## 4.2 CHALLENGE #2: SAMPLE EFFICIENCY AND DATA SCARCITY

Building on the insights from Sections 4.1, we recognize the critical importance of sufficient sampling for accurate estimation of metrics. This section delves into the relationship between sample size and convergence rate for each metric, exploring its impact across various feature spaces. Here, we define convergence rate as the rate at which the distance between training and testing set feature stabilizes as the number of video sample increases. It is a measure of sample efficiency.

Previous studies in the image and audio domains have shown that as the sample sizes $N$ decrease, Fréchet Distances increase (Bińkowski et al., 2021; Gui et al., 2024; Jayasumana et al., 2024; Chong & Forsyth, 2019). This sensitivity to sample size is a common phenomenon among distributional distance metrics: As sample sizes increase, distance metrics become more reliable and accurate. However, while all metrics benefit from additional data, some converge to the true underlying distance more quickly than others.

We investigate the sample size required to achieve convergence within a 5% error margin to average metric distance measured at 5,000 samples. Our comparative analysis in Figure 3 spans two diverse types of datasets: UCF-101, HMDB (human action recognition) and Something-Something-v2 (SSv2, hand gesture recognition). An extended analysis on more datasets, feature spaces, and metrics is found in Appendix E.3. Our key findings include:

1. The sample sizes needed for convergence within the same feature space are similar across datasets. For instance, 4,350 samples on UCF-101 and 4,700 samples on SSv2 are required for FVD (I3D+FD) to converge, while 700 samples on UCF-101 and 800 samples on SSv2 are required for JEDi (MMD$_{POLY}$+V-JEPA$_{SSv2}$) to converge.

2. MMD$_{POLY}$ demonstrates the quickest convergence rate across non-I3D feature spaces. For example, it requires only 1,500 and 700 samples to reach a steady metric value on UCF-101 in V-JEPA$_{PT}$ and V-JEPA$_{SSv2}$ feature spaces, respectively (Figure 14).

3. Fréchet Distance has the worst sample efficiency among metrics, while I3D features exhibit the worst sample efficiency across feature spaces (Figure 14, 16, 15).

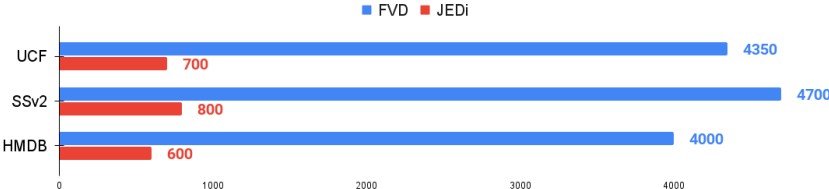

Figure 3: The number of samples required to achieve a 5% error margin in the distance measured from 5,000 samples using the training and testing sets of the 3 listed datasets. We assess the number of samples required for convergence at 100 sample intervals. Convergence at sample size $N$ is achieved if: (1) the average metric value from 5 repeated samplings of $N$ features falls within a 5% error margin, and (2) all subsequent interval evaluations maintain an average metric value within the 5% error margin. The results for other metrics, feature spaces, and different datasets are provided in the Appendix (Figure 14-15). We find that Fréchet Distance (FD) converges slowest, while MMD$_{POLY}$ shows the highest sample efficiency.

As shown in Figure 3, it often takes thousands of video clips for FVD features to converge; however, many datasets contain insufficient amount of unique videos to reach this convergence, making

it challenging to define a robust distribution in such high-dimensional spaces (Pont-Tuset et al., 2018; Geiger et al., 2013; Ebert et al., 2017). This is often worked around by transforming videos into shorter, partly overlapping clips. This method is problematic and biases the metric due to the repetitiveness of the data. This fault has remained largely challenged. For instance, as shown in Figure 16, the BAIR dataset's size and sample efficiency issues are particularly noteworthy. Despite these limitations, BAIR dataset remains a widely-used benchmark for video generation, with numerous studies reporting FVD results on it (Yu et al., 2023; Wu et al., 2021; Voleti et al., 2022).

We note three key hurdles stemming from this sample efficiency issue in video generation: (1) *Data size:* Limited samples compromise estimate reliability, undermining robust statistical analysis; (2) *Computational resources:* Generating samples is computationally expensive and time-demanding; (3) *Metric convergence speed:* Slow convergence rates hinder accurate assessments. While dataset size and computational resources are largely beyond our control, we can address the third concern by selecting metrics with higher sample efficiency where convergence happens with less samples.

## 5 METRIC DISTANCE ANALYSIS: NOISE, GENERATIVE MODELS, AND HUMAN STUDY

This section explores the effects of videos distorted with noise and videos generated at varying model training checkpoints on metric reliability, assessing their impact on: (1) metric accuracy, (2) sample efficiency and (3) human metric alignment.

### 5.1 NOISE & GENERATION MODELS AND THEIR IMPACTS ON METRIC MEASUREMENT

| Metric | No Noise | Blur (low) | Blur (medium) | Blur (high) |
|--------|----------|------------|---------------|-------------|
| FVD | $69.0 \pm 0.187$ | $68.1 \pm 0.087$ | $97.4 \pm 0.151$ | $177.3 \pm 0.221$ |
| JEDi | $0.017 \pm 0.000$ | $0.038 \pm 0.000$ | $0.256 \pm 0.000$ | $0.571 \pm 0.000$ |

Table 1: The table shows average FVD and JEDi distances between training and testing set feature distributions under various blur distortions. The testing video dataset is subjected to noise distortions, including low blur ($\sigma \sim [0.05, 0.75]$), medium blur ($\sigma \sim [0.1, 1.5]$), and high blur ($\sigma \sim [0.01, 3]$), where $\sigma$ represents the per-frame blur intensity, and a larger range indicates greater temporal inconsistency. The experiment is replicated 10 times to account for variability. Our analysis reveals that FVD fails to detect low blur noise and incorrectly suggests an improvement in video quality (highlighted in gray). *Note: To improve readability, we standardize JEDi by applying a scaling factor of 100 to the V-JEPA$_{SSv2}$+MMD polynomial distance.*

This study investigates metric reliability when presented with videos affected by three noise distortion types (salt and pepper noise, temporal blur and elastic distortion) and two image-to-video generation models (I2V-Stable Video Diffusion and Open-Sora).

Salt and pepper noise, a type of impulsive noise, spatially corrupts visual data by randomly altering pixel values to extreme intensities. Elastic noise distortion from (Ge et al., 2024) primarily introduces temporal distortions and occasionally deforms object shapes. In addition, we introduce temporal blur noise which involves applying Gaussian kernels of varying strengths to blur frames, preserving appearance and shape integrity while focusing on temporal distortion. The two generative models we used were adopted from open-source repositories, utilizing the provided checkpoints (Zheng et al., 2024; von Platen et al., 2022). Detailed inference configurations for these models are in Appendix G.

We highlight some of our results in Table 1, and the remaining results are in Figures 17 and 18. The key findings in these experiments include:

1. Metrics in the I3D feature space are impacted by salt and pepper noise (a spatial distortion) significantly more than by other types of distortions. This aligns with the findings of Ge et al., *demonstrating that the I3D feature space is highly sensitive to spatial distortions but less responsive to temporal distortions.* As shown in Section 5.4, I3D does not align with human preferences with respect to distortions.

2. *I3D and VideoMAE$_{PT}$ are not ideal feature spaces for building video quality metrics, as they do not capture blur distortion well.* Notably, they perceive a testing distribution with slight artificial blur

added to the frames as more similar to the training distribution than the original testing distribution. In fact, they estimate the low-blur distorted videos are 10%-20% closer to the ground-truth training distribution.

3. Our experimental evaluation reveals that the choice of feature space have a greater impact on distance values than the choice of distribution distance metric. Moreover, the metrics exhibit consistent performance across diverse datasets, demonstrating robustness.

## 5.2 Metric Robustness Assessment With Progressive Distortion level and Training Duration

**Distortion Level** In Figure 4, we conduct a study to evaluate the performance of metrics in assessing various blur noise distortions on the UCF-101 dataset. All metrics identified the decline in video quality as noise levels increased.

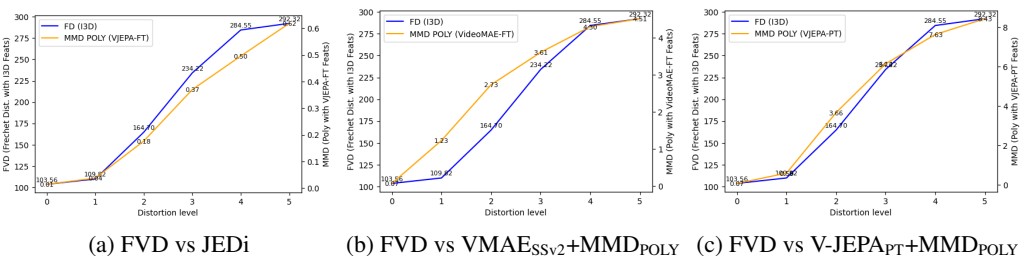

    (a) FVD vs JEDi       (b) FVD vs VMAE$_{SSv2}$+MMD$_{POLY}$   (c) FVD vs V-JEPA$_{PT}$+MMD$_{POLY}$

Figure 4: How metric distance changes as temporal blur increases. Specifically, temporal blur distortion is controlled by varying the sigma range ($\sigma$) using the distortion level ($\lambda$), with $\sigma = [0.1 - 0.01\lambda, 0.75 + 0.8\lambda]$. The study is carried out on the UCF-101 dataset.

**Training Duration** We further aim to evaluate metrics for generative models, focusing on their ability to track changes in video quality throughout training (Figure 5). Due to the computational expense of training video generation models from scratch, we fine-tune Stable Video Diffusion's weights on the BDD dataset using Ctrl-V's code (Luo et al., 2024). Ctrl-V uses a pre-trained SVD model but modifies the input padding strategy to enable multi-frame conditioning. Initially, the visual quality of generated videos is poor, but it improves over the training time. We utilize these fine-tuning steps to assess our metrics' robustness in evaluating fine-tuned model checkpoints. We expect a good metric to decrease steadily over time and thus have a negative correlation close to 1 in magnitude. We visualize several checkpoint generations in Figure 22.

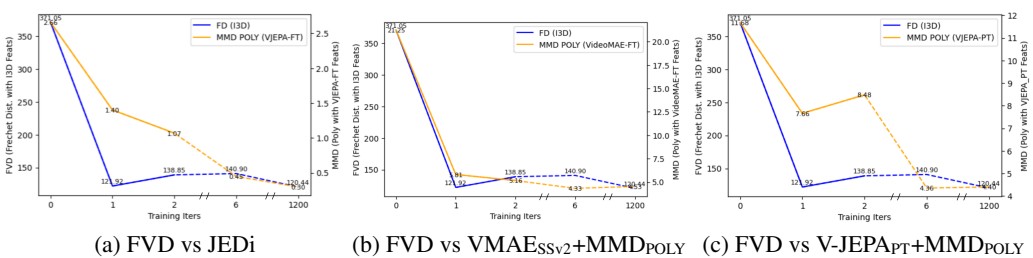

    (a) FVD vs JEDi       (b) FVD vs VMAE$_{SSv2}$+MMD$_{POLY}$   (c) FVD vs V-JEPA$_{PT}$+MMD$_{POLY}$

Figure 5: Ctrl-V is fine-tuned on BDD. Visual inspection show incremental improvements in generation quality at each training step. This is captured by, JEDi (V-JEPA$_{SSv2}$+MMD$_{POLY}$). However, FVD (I3D+FD), VideoMAE$_{SSv2}$+MMD$_{POLY}$ and V-JEPA$_{PT}$+MMD$_{POLY}$ fail to detect incremental improvements. The Spearman coefficient correlation values for the X and Y axes are -1, -0.6, -0.9 and -0.8 for JEDi, FVD, VideoMAE$_{SSv2}$+MMD$_{POLY}$ and V-JEPA$_{PT}$+MMD$_{POLY}$, respectively, with only JEDi showing statistical significance.

**Results** Only JEDi (V-JEPA$_{SSv2}$+MMD$_{POLY}$) successfully tracks incremental gains in all checkpoints, whereas FVD (I3D+FD), VideoMAE$_{SSv2}$+MMD$_{POLY}$ and V-JEPA$_{PT}$+MMD$_{POLY}$ do not.

## 5.3 SAMPLE EFFICIENCY UNDER NOISE DISTORTION

Alongside Section 5.1, we investigate the sample efficiency of various metrics under noisy conditions. Specifically, we measure the number of samples it takes for the distance between the original training distribution and the noise-added testing distribution to stabilize/converge. The noise-added testing set essentially simulates a set of generations. Our findings indicate that: JEDi remains much more sample efficient compared to FVD in this condition. We present our experiment results in Appendix F.2.

## 5.4 HUMAN EVALUATION

To examine human perception of video quality degradation due to noise distortions, we conduct a small-scale survey using 24 randomly selected videos from the UCF-101 and Sky Scene test sets, originally at 30 fps, subsampled to 3-second clips at 7 fps. We apply four noise distortions: two levels of blur, elastic distortion, and salt and pepper noise, with parameters detailed in Appendix F.3. To reduce border effects from elastic distortion, all videos are center-cropped to $230 \times 310$ pixels.

Surveys for the UCF-101 and Sky Scene datasets are conducted randomly, with participants evaluating anonymized video pairs that differ only in noise type. They rate each pair under four comparisons, choosing one video as superior or noting no difference. Each comparison is assessed by 20 independent raters from an academic community, who focus solely on visual quality without knowledge of the datasets or distortions.

The Analytic Hierarchy Process (AHP) Saaty (1987) aggregates responses using a pairwise comparison matrix to derive a priority vector for noise distortion types. This vector is normalized and inverted to align with distribution distance metrics, where 0 represents ideal quality. Scores for each distance metric-feature space combination are normalized and compared to human survey results using cosine similarity. Evaluation results are summarized in Figure 6 and more details can be found in Appendix F.3.

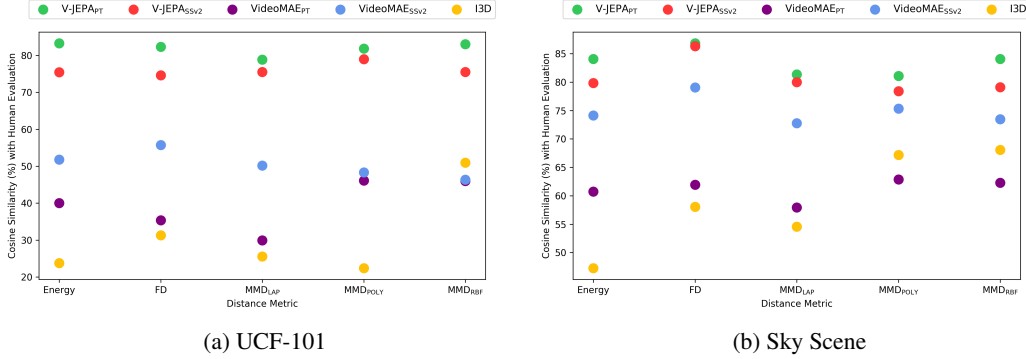

|              (a) UCF-101              |              (b) Sky Scene              |

Figure 6: Alignment of human evaluation with distribution distance metrics. Metrics computed in V-JEPA feature spaces surpass that of both I3D and VideoMAE in terms of alignment with human evaluation.

**Results**    While metrics within a feature space generally perform at the same level, distances calculated in the feature space of V-JEPA$_{SSv2}$ or V-JEPA$_{PT}$ model resoundingly outperform both I3D and VideoMAE-based metrics in terms of alignment with human evaluation. Among raters, there were agreements of 83.70% and 53.54% on the UCF-101 and Sky Scene datasets, respectively. Perhaps intuitively, humans are much more confident assessing content with human activity than with more abstract visuals, where subjective interpretations can vary significantly.

## 5.5 STUDY ON ARTIFICIAL INTELLIGENCE GENERATED CONTENT

To further evaluate our model's performance on Artificial Intelligence Generated Content (AIGC), we leveraged the *Text-to-Video Quality Assessment Database (T2VQA-DB) (Kou et al., 2024)* and *Text and Image Prompt Dataset for Image-to-Video Generation (TIP-I2V) (Wang & Yang, 2024)*.

|  | Tune-A-Video | Show-1 | LaVie |
|---|---|---|---|
| **MOS Rank** | Worst | Median | Best |
| **JEDi (#. conv samples)** | 4.427 (100) | 2.164 (200) | 0.722 (300) |
| **FVD (#. conv samples)** | 884.1 (600) | 445.6 (800) | 279.9 (900) |

Table 2: Quantitative comparison of JEDi and FVD Metrics for T2VQA-DB models, stratified by MOS (Worst, Median, Best). Values represent measured distances; numbers in parentheses indicate sample sizes needed to achieve a 5% error margin (estimated from 1,000 samples).

**T2VQA-DB** comprises 10,000 AI-generated videos by 10 Text-to-Video (T2V) AI models, each associated with a Mean Opinion Score (MOS) reflecting *text-video alignment and fidelity* (Section 3.2 (Kou et al., 2024)). In particular, (1) MOS scores display substantial variance ($\mu = 50, \sigma = 16.6$), (2) model rankings are ambiguous due to score overlap (Figure 3 from (Kou et al., 2024)), and (3) No "ground-truth" video set exists; we addressed this using WebVid10M (Bain et al., 2021) videos with matching prompts.

We assessed the outputs of three models from T2VQA-DB (Tune-A-Video (Wu et al., 2023), Show-1 (Zhang et al., 2024), LaVie (Wang et al., 2023b)) with varying video quality, ranked 10 (worst), 5, and 1 (best) based on MOS scores. Both FVD and JEDi scores ranked these models consistently, with Tune-A-Video as the worst and LaVie as the best. The quantitative findings are presented in Table 2. *Notably, JEDi achieved convergence with significantly fewer samples on T2VQA-DB compared to FVD, requiring only 100, 200, and 300 samples to converge, while FVD required 600, 800, and 900 samples.*

**TIP-I2V** provides the videos generated by Pika (Pika-AI, 2024), SVD (Blattmann et al., 2023) and Open-SORA (Zheng et al., 2024), which we use for our subsequent analysis of the video feature spaces. Specifically, if video sets $A$ and $B$ are from the same distribution, their features extracted using any feature extractor should remain in a same distribution. However, if video sets $A$ and $B$ come from different distributions, a robust feature extractor designed for evaluating distributional differences would highlight these distinctions. The results in Appendix H demonstrate that V-JEPA extracted features reveal clear distributional distinctions between 3 AIGC video sets and 1 real video dataset, whereas I3D features do not exhibit such distinct clustering.

## 6 CONCLUSION

In this study, we carefully look at many aspects of video metrics and find **JEDi** to be the best choice.

**First,** we show that the normality assumption made by the Fréchet Distance (FD) does not hold true in the video feature spaces, and this becomes more evident as the duration of the videos increases. **Second,** we discuss two challenges with FD: 1) Estimating the covariance matrix for FD is difficult due to high latent space dimensionality, and reducing this dimension with autoencoders showed no improvement; 2) The sample efficiency of FD is low; through extensive comparison, we find that Maximum Mean Discrepancy (MMD) with V-JEPA exhibits much higher sample efficiency across all datasets tested. **Third,** we investigate the impact of noise on feature spaces and found that I3D was more sensitive to image quality distortion than temporal distortion, and that I3D and VideoMAE$_{SSv2}$ does not capture blur distortion well. On the other hand, V-JEPA$_{SSv2}$ stands out as the more robust feature space among them. **Fourth,** we observe the correlation between metric with distortion level and training duration. We show that while both FVD (FD+I3D) and JEDi (MMD$_{POLY}$+V-JEPA$_{SSv2}$) are positively correlated with distortion level (higher distance with higher distortion), only JEDi is highly negatively correlated to training duration (lower distance with more training). **Finally,** our human study results indicate that our proposed metric aligns most closely with human preferences. Also, we utilize the AIGC databases with human opinion scores to show that JEDi is a more sample-efficient metric with a superior feature space for analysis.

Based on our comprehensive analysis, JEDi emerges as the most effective and practical metric for guiding the current surge in video generation research. To facilitate the usage of JEDi, we provide simple and easy-to-use code that, given its striking benefits, hope the community will embrace.

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

# A  STATISTICAL DISTRIBUTIONS

## A.1  MULTIVARIATE GAUSSIAN DISTRIBUTION

The probability density function of a $k$-dimensional multivariate Gaussian is given by:

$$P(\mathrm{x}) = \frac{1}{(2\pi)^{k/2}|\Sigma|^{1/2}} \exp\left(-\frac{1}{2}(\mathrm{x}-\mu)^{\mathsf{T}}\Sigma^{-1}(\mathrm{x}-\mu)\right) \tag{2}$$

where $\mathrm{x} \in \mathbb{R}^k$ is the $k$-dimensional random sample, $\mu \in \mathbb{R}^k$ is the mean vector, $\Sigma \in \mathbb{R}^{k \times k}$ is the covariance matrix which is symmetric and positive definite, $|\Sigma|$ denotes the determinant of the covariance matrix, and $\Sigma^{-1}$ denotes the inverse of the covariance matrix.

## A.2  GAUSSIAN MIXTURE MODELS

A Gaussian Mixture Models (GMM) with $c$ clusters is a probabilistic model that assumes the data is generated from a mixture of $c$ Gaussian distributions, each with its own mean and covariance. The probability density function of the GMM is given by: $P(\mathrm{x}) = \sum_{i=1}^{c} \pi_i \mathcal{N}(\mathrm{x}|\mu_i, \Sigma_i)$ where the cluster weights $\pi_i$ sum up to 1. The parameters of GMMs are often estimated using iterative algorithms, such as Expectation-Maximization (EM) algorithm (Dempster et al., 1977), as there is no closed-form solution to maximize its likelihood function.

## A.3  PRELIMINARY: LINEAR TRANSFORMATION OF MULTIVARIATE GAUSSIAN DISTRIBUTION AND LINEAR DIMENSIONALITY REDUCTION METHODS

⌂ Back to paper

Let $\mathrm{x} \sim \mathcal{N}(\mu, \Sigma)$ follow a multivariate Gaussian distribution with mean vector $\mu$ and covariance matrix $\Sigma$. Let $\boldsymbol{A}$ be a matrix and $\boldsymbol{b}$ be a vector. We are interested in the linear transformation $\mathrm{y} = \boldsymbol{A}\mathrm{x} + \boldsymbol{b}$. The following is the proof of y is also a multivariate Gaussian distribution. Specifically, $\mathrm{y} \sim \mathcal{N}(\boldsymbol{A}\mu + b, \boldsymbol{A}\Sigma\boldsymbol{A}^{\mathsf{T}})$.

The moment generating function for x is

$$M_{\mathrm{x}}(t) = \mathbb{E}(\exp[t^T \mathrm{x}]) = \exp\left[t^T\mu + \frac{1}{2}t^T\Sigma t\right] \tag{3}$$

and the moment generating function for y is given by

$$\begin{aligned}
M_{\mathrm{y}}(t) &= \mathbb{E}\left(\exp\left[t^T(\boldsymbol{A}\mathrm{x} + \boldsymbol{b})\right]\right) \\
&= \exp[t^T\boldsymbol{b}]\mathbb{E}\left(\exp[t^T\boldsymbol{A}\mathrm{x}]\right) \\
&= \exp[t^T\boldsymbol{b}]M_{\mathrm{x}}(\boldsymbol{A}^T t) \\
&= \exp\left[t^T(\boldsymbol{A}\mu + \boldsymbol{b}) + \frac{1}{2}t^T(\boldsymbol{A}\Sigma\boldsymbol{A}^T)t\right]
\end{aligned} \tag{4}$$

This indicates that the moment generating function of y aligns with the moment generating function of the multivariate Gaussian distribution. Therefore, y is a random variable that follows a multivariate Gaussian distribution (detailed proof can be found in Soch & et al. (2024)).

Principal Component Analysis (PCA) and Linear Discriminant Analysis (LDA) are statistical methods used to reduce the number of variable dimensions in a dataset. PCA is a process of linear transformation that involves mapping data from a higher dimensional space to a lower dimensional space by identifying the directions in which the data varies the most (F.R.S., 1901). LDA entails a linear transformation that maps data from a higher-dimensional space to a lower-dimensional space in order to effectively separate multi-class objects (Martínez & Kak, 2001). Because PCA and LDA are linear transformations, and we've shown that data from a multivariate Gaussian distribution remains Gaussian after linear transformations, we can conclude that applying PCA or LDA to data from a multivariate Gaussian distribution preserves its Gaussian properties.

## A.4 Distribution Distance Metrics Overview

⌂ Back to paper

*Mixture Wasserstein (MW$_2$) (Delon & Desolneux, 2020):* Optimal transport is a mathematical field that deals with finding a transport plan that minimizes the total cost of moving the mass from the source distribution to the target distribution (Monge, 1781; Montesuma et al., 2023). Recently, Delon & Desolneux (2020) proposed a Wasserstein-type distance within a novel optimal transport framework for Gaussian Mixture Models (GMMs) with restricted couplings. By confining the set of possible coupling measures to GMMs, they derive a simple, discrete formulation of the distance metric, making it computationally efficient for problems with high dimensions. The distance is called Mixture Wasserstein and is denoted as MW$_2$. The MW$_2$ distance is always upper bounded by Wasserstein distance ($W_2$) plus the variances of the Gaussian components. In addition, its computational complexity is solely determined by the number of clusters.

*Energy Statistic (Baringhaus & Franz, 2004; Szekely & Rizzo, 2004):* Energy statistic measures the difference between distributions based on pairwise distances between points. Given $\{x_1, \ldots, x_m\}$ are random samples generated from distribution $P$ and $\{y_1, \ldots, y_n\}$ are random samples generated from distribution $Q$, the energy distance $\mathcal{E}(P, Q)$ is given by:

$$\mathcal{E}(P,Q) = \frac{2}{mn} \sum_{i=1}^{m} \sum_{j=1}^{n} \|x_i - y_j\| - \frac{1}{m^2} \sum_{i=1}^{m} \sum_{j=1}^{m} \|x_i - x_j\| - \frac{1}{n^2} \sum_{i=1}^{n} \sum_{j=1}^{n} \|y_i - y_j\|. \quad (5)$$

The energy statistic is appropriate for comparing complex distributions without making assumptions about a particular underlying distribution.

*Maximum Mean Discrepancy (MMDs) (Gretton et al., 2012):* MMD is a general class of kernel-based sample tests that maximize the mean difference between samples from two distributions by optimizing over all data transformations $f$ within a function space $\mathcal{F}$. Some popular kernel functions used for MMD include: linear, polynomial, sigmoid, Laplace and RBF (Gaussian) kernels.

Given two sets of features, $X = \{\mathbf{x}_1, \mathbf{x}_2, \ldots, \mathbf{x}_m\}$ and $Y = \{\mathbf{y}_1, \mathbf{y}_2, \ldots, \mathbf{y}_n\}$, sampled from $P$ and $Q$, $d_{MMD}^2(P, Q)$ with a given kernel, $k$, is given by (Jayasumana et al., 2024):

$$\tilde{d}_{MMD}^2(X,Y) = \frac{1}{m(m-1)} \sum_{i=1}^{m} \sum_{\substack{j=1 \\ j \neq i}}^{m} k(\mathbf{x}_i, \mathbf{x}_j) + \frac{1}{n(n-1)} \sum_{i=1}^{n} \sum_{\substack{j=1 \\ j \neq i}}^{n} k(\mathbf{y}_i, \mathbf{y}_j) - \frac{2}{mn} \sum_{i=1}^{m} \sum_{j=1}^{n} k(\mathbf{x}_i, \mathbf{y}_j).$$

$$(6)$$

Like the energy statistic, MMD is distribution-free, requiring no assumptions about the underlying distributions of $P$ or $Q$.

## B Computation Configurations

### B.1 Feature Extractors

Feature extraction is performed on a *single NVIDIA RTX 4080 GPU with float32 precision*. However, VideoMAE-v2 features require a more specialized setup: *a single NVIDIA RTX A100 GPU with 80G memory*. Notably, VideoMAE-v2 precision varies by clip length: *float32 for clips under 64 frames and float16 for longer clips*. We use batch sizes of 10 (clips < 64 frames) and 2 (clips ≥ 64 frames) for feature extraction.

**I3D Configuration**[2] We adopt the recommended feature extractor from the FVD paper (Unterthiner et al., 2019): I3D logits features pre-trained on Kinetics-400.

**VideoMAE$_{PT}$ and VideoMAE$_{SSv2}$** We compute VideoMAE features using the official PyTorch implementation (Wang et al., 2023a), following the guidelines outlined in Ge et al..

---

[2]Feature extractor for FVD

**V-JEPA$_{PT}$ Configuration** We compute V-JEPA features using the official PyTorch implementation (Bardes et al., 2024). Specifically, our V-JEPA pretrained model, referred to as V-JEPA$_{PT}$, consists solely of the V-JPEA encoder. Consistent with previous research (Ge et al., 2024), we extract features from the encoder and average them across all feature tokens.

**V-JEPA$_{SSv2}$ Configuration**[3] We compute V-JEPA features using the official PyTorch implementation (Bardes et al., 2024). Specifically, our V-JEPA classifier model, referred to as V-JEPA SSv2 fine-tuned model or V-JEPA$_{SSv2}$, consists of two components: (1) a V-JEPA encoder, and (2) an adaptive probe with attentive pooler. Consistent with previous research (Ge et al., 2024), we exploit the pre-logit features generated by the attentive pooler.

### B.2 DISTRIBUTION METRIC CONFIGURATIONS

**Fréchet Distance**[4] We calculate the Fréchet Distance using torchaudio's functional API, passing mean and covariance statistics to `frechet_distance`.

**Energy statistic** We utilize the `https://github.com/josipd/torch-two-sample/` repository to calculate the energy statistic.

**Mixture Wasserstein** We utilized the official implementation from Delon & Desolneux's repository to calculate the Mixture Wasserstein distance.

**Mean Maximum Discrepancy** We compute MMD distances using code from (Wang et al.), with kernel-specific parameters:

- **RBF/Laplacian:** $\gamma = 1/\text{ndim}$
- **Polynomial**[5]**:** degree = 2, $\gamma = 1$, coef = 0.

## C FRÉCHET VIDEO DISTANCE: A SEMI-METRIC

A distance metric on a set $\mathbb{X}$ is a function $d : \mathbb{X} \times \mathbb{X} \to \mathbb{R}$ that satisfies the following properties for all points $x, y, z \in \mathbb{X}$: **1.** *Non-negativity:* $d(x, y) \geq 0$; **2.** *Identity of indiscernible:* $d(x, y) = 0 \iff x = y$; **3.** *Symmetry:* $d(x, y) = d(y, x)$; and **4.** *Triangle inequality:* $d(x, z) \leq d(x, y) + d(y + z)$.

Fréchet Distance (FD) satisfies all the properties of a metric, except for the triangle inequality, which establishes it as a semi-metric. The triangle inequality is a crucial property that underpins the linearity of a metric and offers valuable interpretability. We will explore this aspect in greater detail later. For now, we will demonstrate the proof for FD being a semi-metric.

The FD of multivariate Gaussian distributions, represented by Equation 1, is composed of two terms: the squared-Euclidean distance between the mean vectors and a term involving the covariance matrices. In order to demonstrate that FD is a semi-metric, we will examine these two components individually.

The first-term, $(\mu_P - \mu_Q)^2$: By definition, the squared Euclidean distance between the mean vectors is non-negative and equals zero if and only if the mean vectors are identical, i.e., $(\mu_P - \mu_Q)^2 \geq 0$ with equality iff $\mu_P = \mu_Q$. Also, the squared Euclidean distance is a symmetric operation. Thus, it satisfies the first three properties. However, the triangle inequality is not satisfied. A counter-example to prove this is: let $\mu_A = [0, 0]^\intercal, \mu_B = [1, 1]^\intercal, \mu_C = [5, 5]^\intercal$, and $(\mu_A - \mu_C)^2 = 50 \geq (\mu_A - \mu_B)^2 + (\mu_B - \mu_C)^2 = 34$.

The second-term, $\text{Tr}\left(\Sigma_P + \Sigma_Q - 2(\Sigma_P \Sigma_Q)^{\frac{1}{2}}\right)$: According to Dowson & Landau (1982), the square root of the second term is considered a natural metric on the space of real covariance matrices of a given order. This implies that the first three properties should hold. Below are the corresponding

---

[3]Feature extractor for JEDi

[4]Distribution metric for FVD

[5]Distribution metric for JEDi

proofs:

$$\text{Tr}\left(\frac{\Sigma_P + \Sigma_Q}{2}\right) \geq \text{Tr}\left(\sqrt{\Sigma_P \Sigma_Q}\right) \quad \text{Arithmetic mean is greater than or equal to geometric mean.}$$

$$\Sigma_P = \Sigma_Q \Rightarrow \text{Tr}\left(\frac{\Sigma_P + \Sigma_P}{2}\right) = \text{Tr}\left(\sqrt{\Sigma_P \Sigma_P}\right) = \text{Tr}(\Sigma_P) \quad \text{If the covariance matrices are the same, the 2nd-term becomes 0.}$$

$$\text{Tr}\left(\frac{\Sigma_P + \Sigma_Q}{2}\right) = \text{Tr}\left(\sqrt{\Sigma_P \Sigma_Q}\right) \Rightarrow \Sigma_P = \Sigma_Q \quad \text{Arithemetic mean equals geometric mean when covariance matrices are identical.}$$

$$(7)$$

To illustrate the symmetric property of the second term, let's assume $\mu_P = \mu_Q = \mathbf{0}$ for simplicity. Let $\mathbf{X}$ represent column vectors sampled from a normal distribution with mean $\mu_P$ and covariance $\Sigma_P$, and $\mathbf{Y}$ represent column vectors sampled from a normal distribution with mean $\mu_Q$ and covariance $\Sigma_Q$. There exists a linear transformation $\Gamma$ such that $\mathbf{Y} = \Gamma\mathbf{X}$. According to Chafaï (2010), $D_{\text{Fréchet}}^2(P, Q)$ can be derived as:

$$D_{\text{Fréchet}}^2(P, Q) = \text{Tr}(\Sigma_P) + \text{Tr}(\Sigma_Q) - \mathbb{E}(\langle \mathbf{X}, \Gamma\mathbf{X}\rangle) \tag{8}$$

To prove the symmetric property, we only need to demonstrate that $\mathbb{E}(\langle \mathbf{X}, \Gamma\mathbf{X}\rangle) = \mathbb{E}(\langle \Gamma\mathbf{X}, \mathbf{X}\rangle)$ because the remaining terms are symmetric. Below is the corresponding proof:

$$\mathbb{E}(\langle \mathbf{X}, \Gamma\mathbf{X}\rangle) = \mathbb{E}(\mathbf{X}(\Gamma\mathbf{X})^\intercal) = \mathbb{E}(\mathbf{X}\mathbf{X}^\intercal\Gamma^\intercal) = \Sigma_P\Gamma^\intercal, \qquad \mathbb{E}(\langle \Gamma\mathbf{X}, \mathbf{X}\rangle) = \mathbb{E}(\Gamma\mathbf{X}\mathbf{X}^\intercal) = \Gamma\Sigma_P$$

$$\Sigma_P\Gamma^\intercal = \Sigma_P^\intercal\Gamma^\intercal = \Gamma\Sigma_P \Rightarrow \mathbb{E}(\langle \mathbf{X}, \Gamma\mathbf{X}\rangle) = \mathbb{E}(\langle \Gamma\mathbf{X}, \mathbf{X}\rangle) \quad \text{Covariance matrices are symmetric: } \Sigma_P = \Sigma_P^\intercal$$

$$(9)$$

Lastly, the triangle inequality is not satisfied. A counter-example to prove this is: let $\Sigma_A = \left(\begin{smallmatrix} 1 & 0 \\ 0 & 1 \end{smallmatrix}\right)$, $\Sigma_B = \left(\begin{smallmatrix} 4 & 0 \\ 0 & 4 \end{smallmatrix}\right)$, $\Sigma_C = \left(\begin{smallmatrix} 9 & 0 \\ 0 & 9 \end{smallmatrix}\right)$,

$$\text{Tr}(\Sigma_A + \Sigma_C - 2(\Sigma_A\Sigma_C)^{\frac{1}{2}}) = 8 \geq \text{Tr}(\Sigma_A + \Sigma_B - 2(\Sigma_A\Sigma_B)^{\frac{1}{2}}) + \text{Tr}(\Sigma_B + \Sigma_C - 2(\Sigma_B\Sigma_C)^{\frac{1}{2}}) = 4$$

When combining the first and second terms with addition, their mathematical properties still hold because their input parameters are different and do not affect each other.

The triangle inequality property is crucial, as it ensures that the distance between two points remains consistent and intuitive. In video generation, models aim to learn the underlying patterns of a real data distribution ($R$) by training on an empirical dataset ($R_{\text{empirical}}$). This involves understanding a probability distribution over potential videos, allowing newly generated videos to closely resemble the structure and content of the observed data. Note that the empirical distribution is an approximation, potentially biased towards the specific sample. The discrepancy between the true distribution and empirical distribution can be quantified using metrics, such as:

$$R_{\text{empirical}} \xrightarrow{N\to\infty} R \implies D(R_{\text{empirical}}, R) \xrightarrow{N\to\infty} 0 \tag{10}$$

where $N$ is the number of samples. The equation demonstrates that as the sample size approaches infinity, the discrepancy between the empirical and true distributions converges to zero. However, in practice, most datasets are finite, and training is typically done on a limited number of samples. As a result, the difference between the empirical and true distributions is bounded by a specific value, rather than reaching zero. Triangle inequality provides insight into the upper-bound of the model generation quality based on the true distribution, expressed as $D(R_{\text{empirical}}, R) + D(G, R_{\text{empirical}}) \geq D(G, R)$.

On the other hand, much of the video generation work involves using pre-trained models to perform zero-shot inference on different datasets in order to test the models' performance and domain adaptation abilities (Hong et al., 2022; Singer et al., 2022; Blattmann et al., 2023; Zhou et al., 2023). However, measuring a model's generation quality on another dataset requires setting up the model for the new dataset and generating numerous samples, which demands significant time, effort, and computational resources. If a distribution distance metric follows the triangle inequality and one only wants to compute an upper bound to validate a model's generation quality on another dataset, it is sufficient to compute: $D(R_{\text{empirical}}^{\text{dataA}}, G) + D(R_{\text{empirical}}^{\text{dataA}}, R_{\text{empirical}}^{\text{dataB}}) \geq D(G, R_{\text{empirical}}^{\text{dataB}})$.

Finally, and most importantly, triangle inequality provides valuable insights into the magnitude of metric distances. Consider the example where $D(G_{\text{modelA}}, R_{\text{empirical}}) = 2D(G_{\text{modelB}}, R_{\text{empirical}})$. With the triangle inequality constraint, we can deduce that model A's performance is at least twice as suboptimal as model B's. Without this constraint, we could only conclude that model A underperforms model B, but without a precise measure of the extent of this underperformance.

# D    NON-GAUSSIAN CHARACTERISTICS OF I3D FEATURES

⌂ Back to paper

## D.1    VISUALIZATION IN LOWER DIMENSIONALITY

Figure 7 displays 2D PCA and LDA projections of I3D features, highlighting non-Gaussian characteristics within individual datasets. The numbers listed in the first column represent the quantity of frames per clip utilized for the experiments in the respective row. In each experiment, we randomly selected 5,000 video clips from each dataset and obtained their I3D features. We performed principal component analysis on each set of 5,000 samples, and the red point clouds in columns 2-5 represent the I3D features projected onto their first 2 principal components.

The point clouds displayed in the last two columns depict the LDA dimensionally-reduced I3D features using the classification labels from the dataset. The markers' colors represent the class labels in the dataset. In each figure, we superimpose blue contour lines on the plot, which delineate the ellipsoidal contours of a 2D Gaussian distribution. Specifically, these contours are computed directly from the mean and covariance matrices of the 2D point clouds, providing a visual representation of the multivariate Gaussian distribution suggested by the Fréchet Video Distance (FVD) metric. We conducted a similar analysis for V-JEPA$_{PT}$ and V-JEPA$_{SSv2}$ features, as shown in Figure 8 and Figure 9.

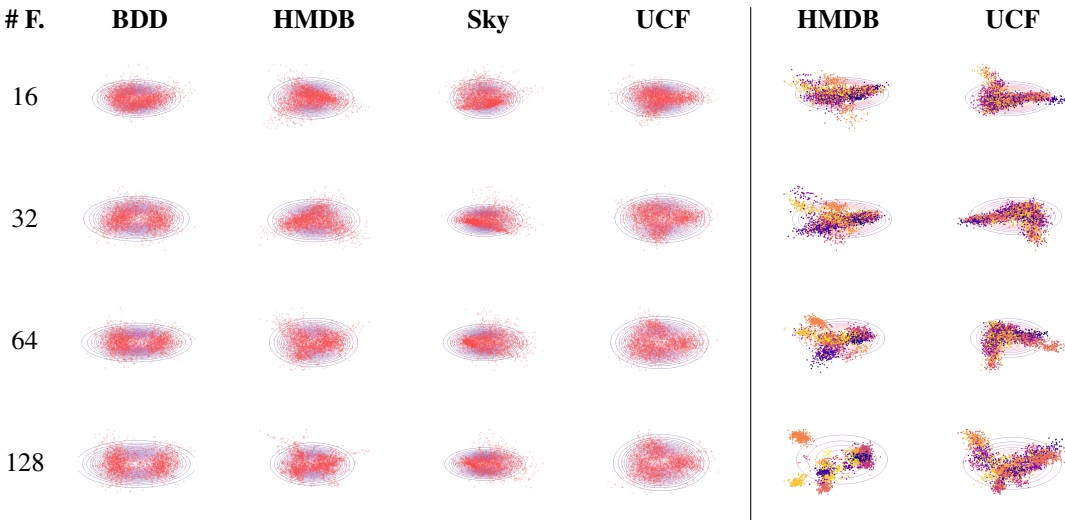

Figure 7: This figure provides empirical evidence of non-Gaussianity in I3D feature space across individual datasets.
⌂ Back to paper

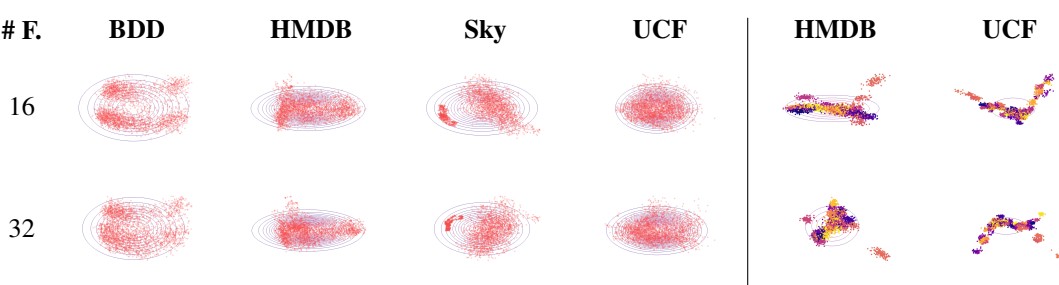

Figure 8: This figure provides empirical evidence of non-Gaussianity in V-JEPA$_{PT}$ feature space across individual datasets.
⌂ Back to paper

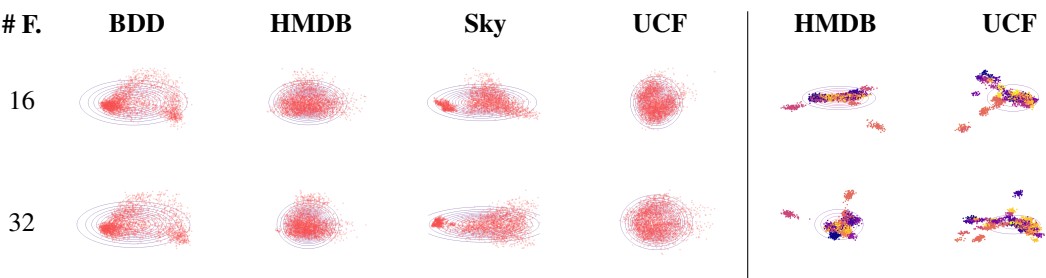

Figure 9: This figure provides empirical evidence of non-Gaussianity in V-JEPA$_{\text{SSv2}}$ feature space across individual datasets.

### D.2  Multivariate Normality Tests

The notion that I3D video features do not follow multivariate Gaussian distributions was further investigated using Mardia's Skewness (Mardia, 1970), Mardia's Kurtosis (Mardia, 1970), and the Henze-Zirkler (Henze & Zirkler, 1990) normality tests, following (Jayasumana et al., 2024). The analysis was done on 16-, 32-, and 128-frame versions of the Anime-Run-v2, BAIR, BDD100k, DAVIS, Fashion Modeling, HMDB-51, How2Sign, KITTI, Something-Something-v2, Sky Scene (Xiong et al., 2018), and UCF-101 datasets, as well as on a baseline dataset constructed by sampling from a multivariate Gaussian distribution with 100 features.

The null hypothesis that the distributions of I3D features were drawn from a multivariate Gaussian distribution was rejected for each of the datasets and normality tests. All three tests accepted the null hypothesis for the multivariate Gaussian baseline dataset.

## E  Experimental Evaluation: Convergence Rates of Distributional Metrics and Dimensionality Reduction Methods

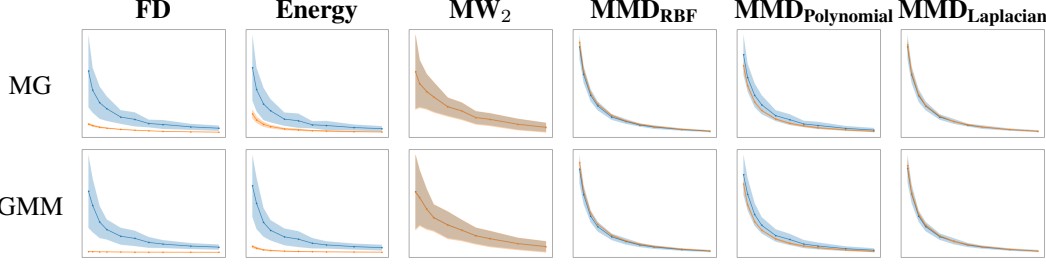

Figure 10: The figures illustrate the evolution of distance estimates between two identical distributions as a function of sample size. The x-axes represent the number of samples drawn from each distribution, while the y-axes corresponds to the distance measurements. The plots in the top-row experiments employ a 100-dimensional multivariate Gaussian distribution, denoted as MG. In this distribution, the first 50 dimensions follow a standard normal distribution ($\mu = \mathbf{0}, \Sigma = I$), while the remaining 50 dimensions are generated as the cumulative sum of the first 50 dimensions, resulting in a structured correlation pattern. In contrast, the bottom-row plots utilize a 100-dimensional Gaussian mixture model comprising 5 clusters, mirroring the multivariate Gaussian setup, with the first 50 dimensions drawn from a GMM and the last 50 dimensions representing the cumulative sum of the first 50 dimensions. The *blue* lines represent the metrics calculated using samples directly drawn from the original distributions, while the *orange* lines represent the metrics computed using samples that have undergone PCA-based dimensionality reduction to 50 principal components. Since the compared distributions are identical, the ideal estimated distance between them should be zero.

⌂ Back to paper

## E.1 TOY EXPERIMENT: METRIC CONVERGENCE RATE

⌂ Back to paper

This section reports on simulated experiments analyzing two key questions: (1) How do different metrics converge? and (2) What role does dimensionality play in shaping convergence rates?

Figure 10 illustrates the findings of a series of simulated experiments examining the empirical distribution distances across various metrics, uncovering the intricate interplay between sample sizes, dimensionality-reduced features, and metric convergence rates. The experimental results shown in the figures demonstrate two key findings: (1) the accuracy of the metrics improves consistently as the empirical sample size increases, validating the proof of concept; and (2) dimension-reduced data can significantly enhance the convergence rate and accuracy of certain metrics (notably FD and energy), as evidenced by the faster convergence of the orange curves relative to the blue curves.

Our simulation experiments also yielded the notable observation that kernel-based distance metrics for distributions exhibit reduced sensitivity to data dimensionality, attributable to their inherent feature mapping, reliance on pairwise distance calculations, and regularization properties inherent in kernel selection. This diminished sensitivity confers a significant advantage in mitigating the curse of dimensionality, thereby enhancing the robustness and reliability of distance metric estimates in higher dimensional space.

In contrast, GMMOT involves fitting separate Gaussian Mixture Models (GMMs) to the sample sets and then computing the optimal transport distance between the fitted GMMs. In the GMM fitting process, the Expectation-Maximization (EM) algorithm updates the parameters of the Gaussian mixtures. During the maximization step, cluster means and covariances are computed using the sample responsibilities calculated in the expectation step. Notably, the sum of the weighted responsibilities for each cluster is less than the total number of samples, but the number of parameters for each cluster is the same as a single multivariate Gaussian distribution. As a result, because of the smaller effective sample size, it is necessary to obtain more samples in order to accurately fit a GMM. Therefore, Mixture Wasserstein ($MW_2$) has the slowest rate of convergence compared to all other metrics, leading us to exclude it from our subsequent video analysis.

## E.2 AUTOENCODERS: MODELS AND TRAINING SPECIFICATION

⌂ Back to paper

In our study, we trained specialized autoencoders for each video representation space to account for variations in video length. Notably, we observed a representation shift with differing video lengths: as shown in Figures 7 and 8, features extracted from videos with different lengths have visible differences; features from 16f and 32f videos are more similar to each other while those from 64f and 128f videos are similar to each other. To address this, we divide the videos from 11 different datasets (UCF, HMDB, etc) into two groups: short clips (16–32 frames) and long clips (64–128 frames). For each group, we extract up to 10,000 features using five different feature extractors (e.g., I3D, V-JEPA$_{PT}$). These results in separate collections of features for short and long clips across all datasets. For instance, I3D yields up to 110,000 short-clip features and up to 110,000 long-clip features from the 11 datasets. We then train two autoencoders for each feature extractor: one using the short-clip features and one using the long-clip features. This gives us a total of 10

autoencoders. The autoencoders' architectures are specified in Algorithm 1 and Algorithm 2.

**Algorithm 1:** I3D Autoencoder Configuration

```
input : in_dim=400

encoder = Sequential(
   Linear(in_dim, in_dim//2),
   ReLU(),
   Linear(in_dim//2, in_dim//4),
   ReLU(),
   Linear(in_dim//4, in_dim//6),
)
decoder = Sequential(
   Linear(in_dim//6, in_dim//4),
   ReLU(),
   Linear(in_dim//4, in_dim//2),
   ReLU(),
   Linear(in_dim//2, in_dim),
)
```

**Algorithm 2:** VideoMAE and V-JEPA Autoencoder Configuration

```
input : in_dim=1408 if VideoMAE else 1280

encoder = Sequential(
   Linear(in_dim, in_dim//3),
   ReLU(),
   Linear(in_dim//3, in_dim//4),
   ReLU(),
   Linear(in_dim//4, in_dim//8),
)
decoder = Sequential(
   Linear(in_dim//8, in_dim//4),
   ReLU(),
   Linear(in_dim//4, in_dim//3),
   ReLU(),
   Linear(in_dim//3, in_dim),
)
```

### E.3 SAMPLE CONVERGENCE ANALYSIS

⌂ Back to paper

Figures included in this section are:

**Figure 11** The evolution of UCF-101 train-test distances for all metrics in all feature spaces.

**Figure 12** A comparison of convergence rates of FVD and JEDi, comparing training and testing sets on UCF101 and Something-Somethingv2.

**Figure 13** Convergence rates UCF-101 train-test distances in VideoMAE$_{PT}$ and V-JEPA$_{PT}$ feature spaces.

**Figure 14** Visualization of the sample size required for VideoMAE$_{PT}$ and V-JEPA$_{PT}$ features to converge to a 5% error margin compared to the distance measured from 5,000 samples using the train and testing sets of UCF-101 and SSv2.

**Figure 15** Visualization of the sample size required for V-JEPA$_{PT}$ and V-JEPA$_{PT}$ features to converge to a 5% error margin compared to the distance measured using the train and testing sets of 9 other datasets. The pink vertical lines in the plots denote the sample size used to compute the target metric distance (ideally 5,000 samples). Note in Figures 3 and 14 that UCF-101 and SSv2 have sufficient samples ($> 5,000$) in their training and testing sets. However, many datasets in this graph have fewer samples. Importantly, convergence estimates become less reliable as bars approach the pink line, due to fewer iterations meeting the second convergence criterion (referring to Figure 3's caption).

**Figure 16** The BAIR dataset (Ebert et al., 2017) demonstrates a perfect example of the convergence issue due to insufficient samples. With 250 training videos, the estimated sample size for convergence (bars) nearly coincides with the target metric computation sample size (pink line). As convergence estimates degrade near this threshold due to insufficient iterations meeting the second criterion, it is difficult to confirm whether metrics truly converge at the displayed sample sizes, especially for Fréchet Distance-based metrics.

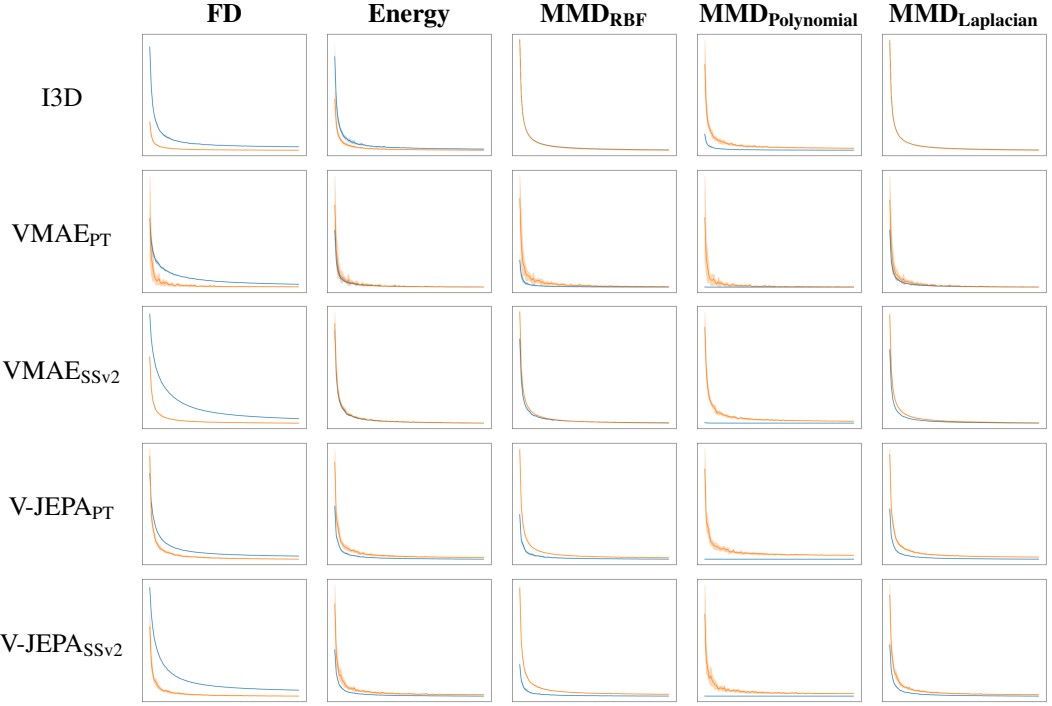

Figure 11: The figures depict the changes in distance estimates between the training and testing sets of the UCF-101 dataset. We extract features from video clips of 32-frame duration using the models indicated in the left-most column. The x-axes represent the number of samples drawn from each distribution, while the y-axes corresponds to the distance measurements. We repeat each experiment 10 times. The lighter shaded area on the plots indicate the variance across these 10 runs. The *blue* lines represent the metrics calculated using features directly extracted from the models, while the *orange* lines represent the metrics computed using features that have been compressed by the autoencoder's encoder. The autoencoder's structure is described in Appendix E.2.
⌂ Back to paper

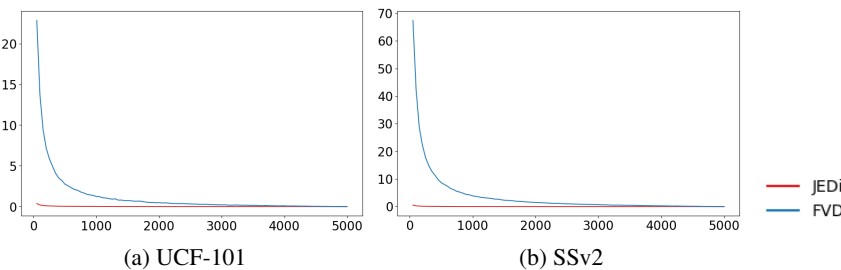

(a) UCF-101            (b) SSv2

Figure 12: A comparison of convergence rates of FVD and JEDi, comparing training and testing sets on UCF101 and Something-Somethingv2. We evaluate convergence rate at 100 intervals, from 50 to 5,000 samples, with 50-sample increments. The x-axes represent the number of samples drawn from the training and test distributions, while the y-axes show the convergence rate, calculated as $\frac{\bar{D}_{\mathrm{m}}(n) - \bar{D}_{\mathrm{m}}(5000)}{\bar{D}_{\mathrm{m}}(5000) + \epsilon}$ where $\epsilon$ is an arbitrarily small number and m is drawn from a set of metrics $\mathcal{M}$. Our methodology involves sampling $n$ samples from the training and testing sets 10 times, computing the distance 10 times for each sampled sets, and calculating $\bar{D}(n)$ as the mean distance across the 10 runs.
⌂ Back to paper

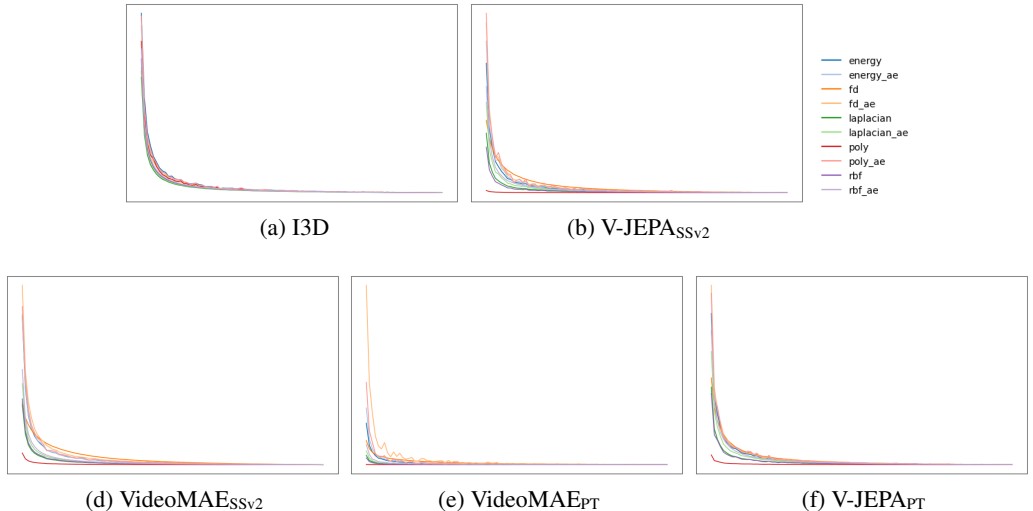

Figure 13: Convergence rates of distributional metrics on UCF-101, comparing training and testing sets in various feature spaces. The convergence rate computation in these figures follows the same configuration as Figure 12, with the x-axis representing the number of samples and the y-axis showing the convergence rate.

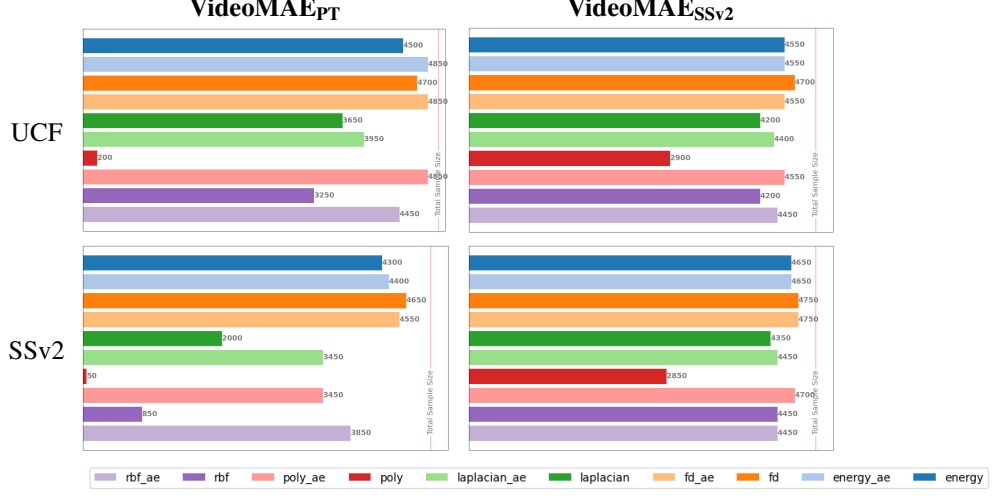

Figure 14: This figure shows the number of samples needed for VideoMAE$_{PT}$ and VideoMAE$_{SSv2}$ to achieve a 5% error margin of the distance measured from 5,000 samples using the training and testing sets. An "_ae" suffix indicates that the feature space has been compressed using an autoencoder. The convergence requirement is stated in Figure 3.

⌂ Back to paper

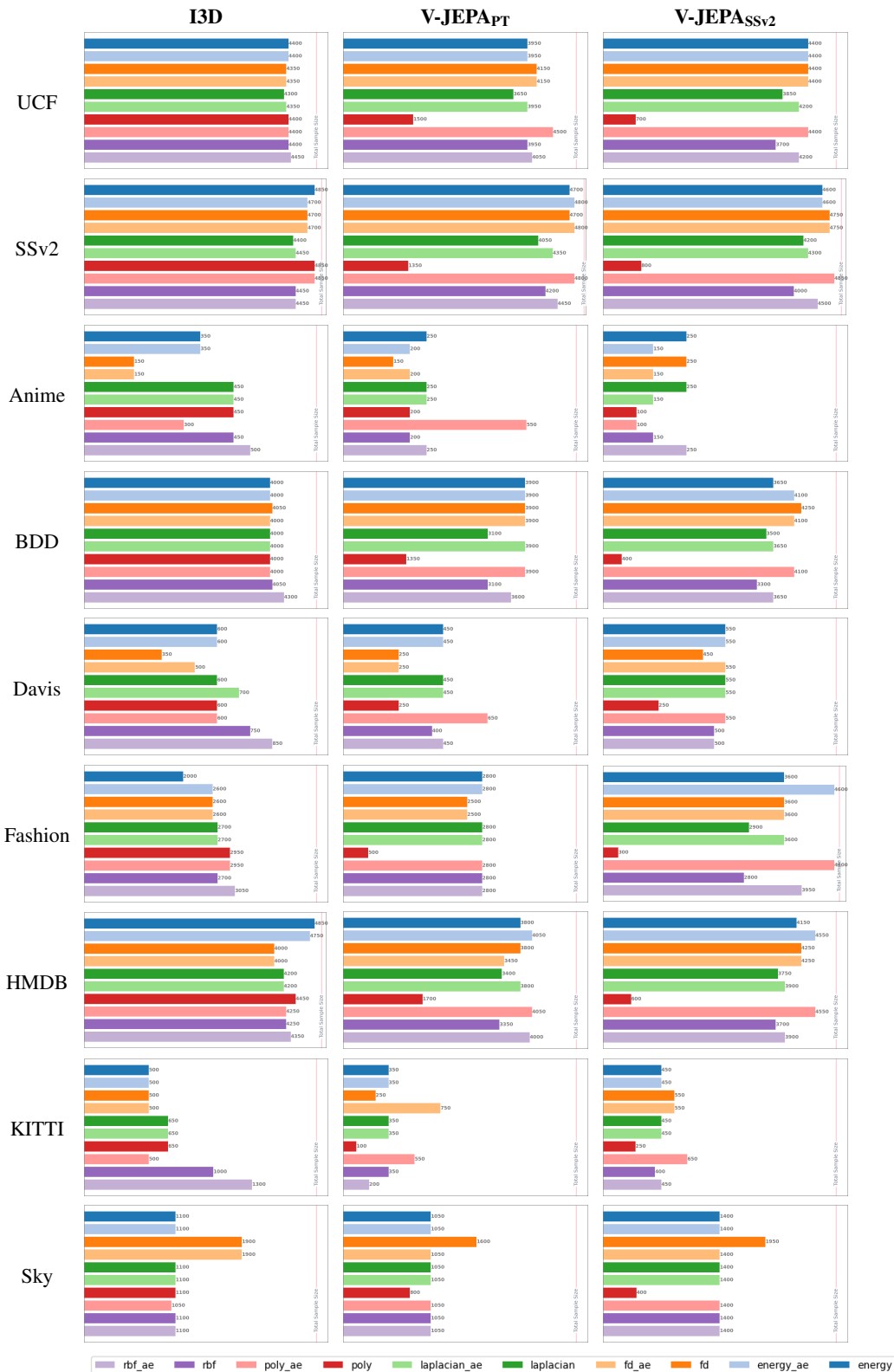

Figure 15: This figure shows the number of samples needed for V-JEPA$_{PT}$ and V-JEPA$_{SSv2}$ to achieve a 5% error margin of the distance measured from 5,000 samples using the training and testing sets on most of the datasets presented in this study. An "_ae" suffix indicates that the feature space has been compressed using an autoencoder. The convergence requirement is stated in Figure 3.

⌂ Back to paper

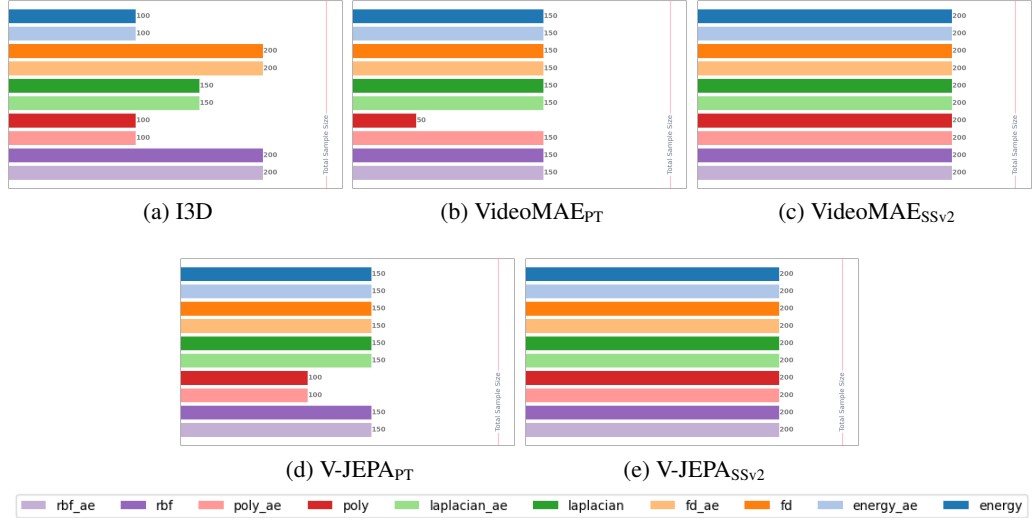

Figure 16: The figures display convergence rates of distributional metrics on BAIR, comparing training and testing sets in VideoMAE$_{PT}$ and V-JEPA$_{SSv2}$ feature spaces.
⌂ Back to paper

## F NOISE DISTORTION STUDIES

### F.1 COMPLIMENTARY MATERIAL FOR THE NOISE AND GENERATIVE MODEL STUDY

Figure 18 illustrates the impact of noise and generative models on the metrics in VideoMAEs and V-JEPAs spaces. The study demonstrates a distinction in how different feature spaces rank these noise types. In Section 5.4, the findings from a human survey to determine which model has the closest ranking with human perception are reported.

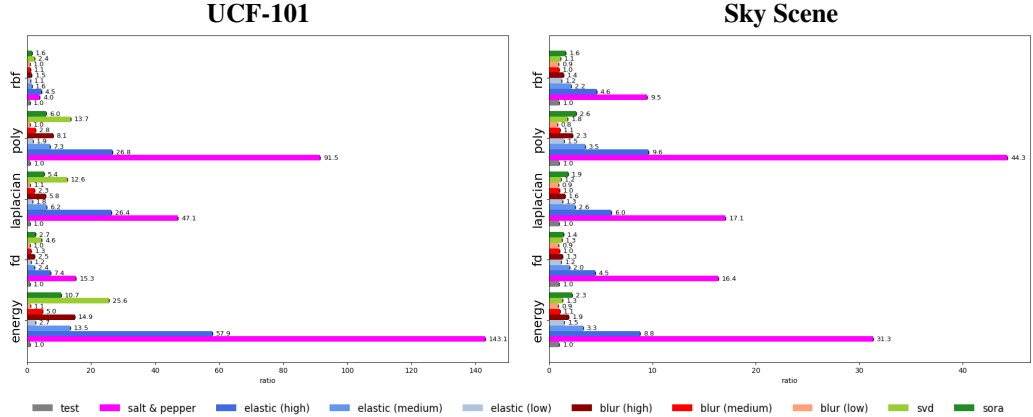

Figure 17: Comparing the FVD (I3D+FD) between training and test sets, using 5,000 samples from each, under various conditions, including noise and conditional generation. The train-test distance (gray) serves as a baseline for evaluating the reliability of the metric. For clarity, we normalize metric values using the train-test distance. The displayed bar values represent these scaled distances. *Notably, the Sky Scene (Xiong et al., 2018) experiment shows that low blur distortion brings the test distribution closer to the training distribution, highlighting a flaw in the FVD metric.*
⌂ Back to paper

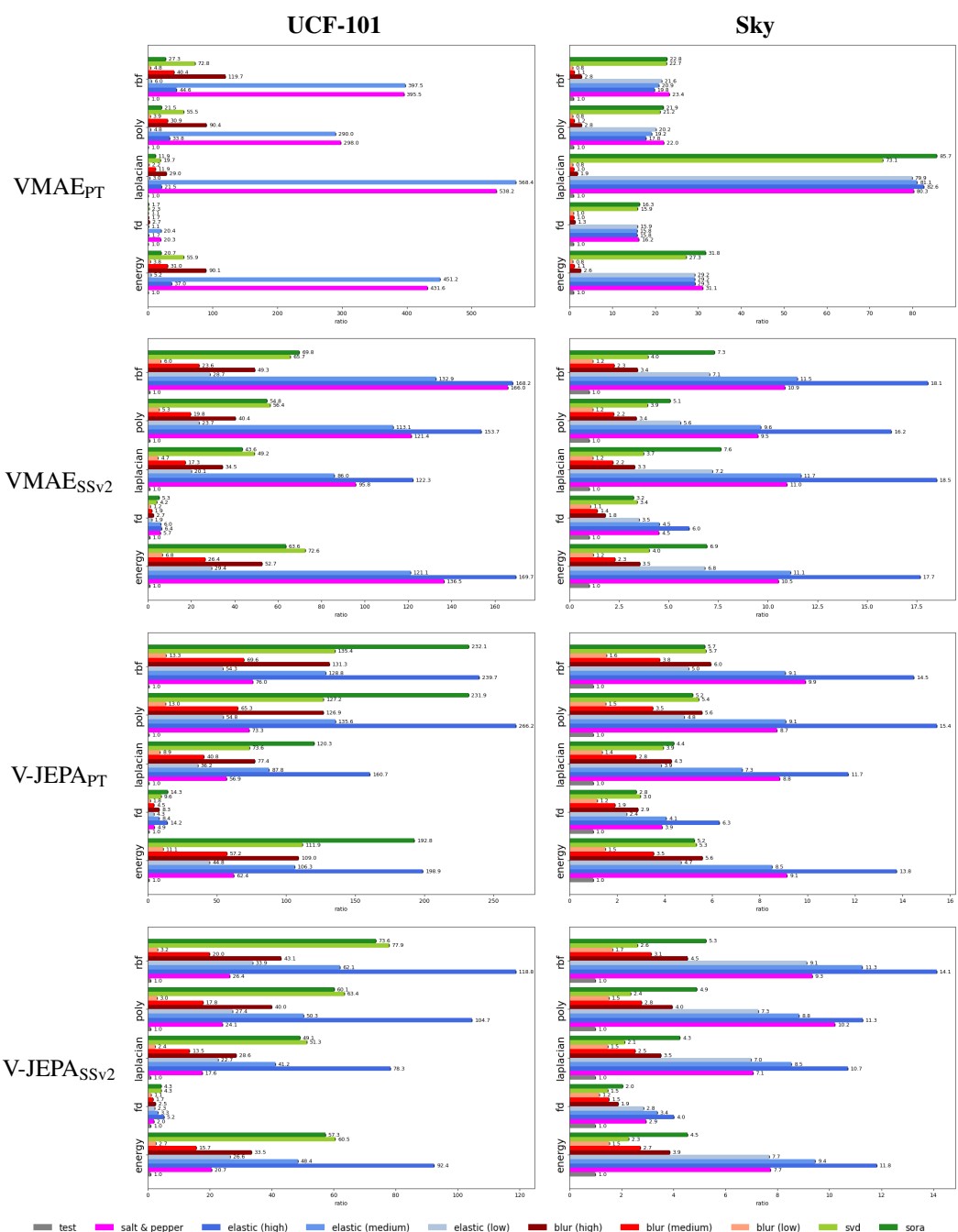

Figure 18: Experiments were conducted on the feature spaces of VideoMAE and V-JEPA using configurations analogous to those in Figure 17. The train-test distance (gray) serves as a baseline for evaluating the reliability of the metric. For clarity, we normalize metric values using the train-test distance. The displayed bar values represent these scaled distances. Both the V-JEPA$_{PT}$ and VideoMAE$_{PT}$ implementations require averaging across patch embeddings. Intuitively, V-JEPA$_{PT}$ metrics are less affected by pixel-level salt and pepper noise as its training is done in an abstract representation space, in direct contrast to that of a VideoMAE.

⌂ Back to paper

## F.2 NOISE AND CONVERGENCE RATE

⌂ Back to paper

Our study investigates the impact of noise and generative model outputs on the convergence speed of FVD and our proposed metric. Figure 19 reveals two key findings:

1. Our method exhibits significantly faster convergence than FVD even with added noise.
2. The presence of noise surprisingly accelerates convergence for both metrics.

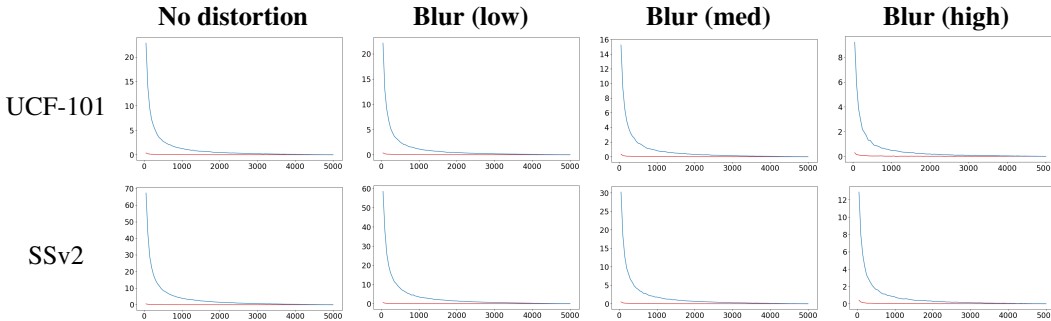

Figure 19: The convergence speed of FVD and the proposed metric is evaluated with blur distortion noises on the UCF-101 and Something-Something-v2 datasets. The convergence rate computation follows the same configuration as Figure 12, with the x-axis represnting th enumber of samples and the y-axis showing the convergence rate. The blue line in the figure represents the convergence rate of FVD, while the red line represents the convergence rate of the proposed metric.

## F.3 HUMAN EVALUATION

⌂ Back to paper

To investigate human alignment on the perception of video quality degradation under various noise distortions, we conduct a small scale survey. The rater population consists of 75% males, 25% females, with 55% under the age of 25, 35% between the ages of 25-35, and 10% above the age of 50. We randomly select 24 videos from each of the UCF-101 and Sky Scene test sets, originally captured at 30 frames per second (fps), and subsample them to 25 frames at 7 fps to generate clips three seconds in length. Four types of noise distortions are systematically applied: high blur with a 7x7 kernel and a Gaussian standard deviation ranging from 0.01 to 3, medium blur with the same 7x7 kernel and a standard deviation ranging from 0.1 to 1.5, elastic distortion with a deformation strength of 30, and salt-and-pepper noise applied at a rate of 1%.

Following the Analytic Hierarchy Process (AHP) (Saaty, 1987), a pairwise comparison matrix is used to aggregate the responses. It is important to note that the sums of corresponding comparisons may not total 100% due to the inclusion of an option indicating no discernible difference in quality between two videos (i.e., all self-to-self comparisons yield a value of 0%). To account for this, the columns are normalized before computing the priority vector, which captures human preference over the different types of noise included in our study. Raw metric values can be found in Appendix F.1.

| Noise Type | Blur (high) | Blur (medium) | Elastic (medium) | Salt and pepper |
|---|---|---|---|---|
| Blur (high) | 0.00% | 0.00% | 12.50% | 2.50% |
| Blur (med) | 93.75% | 0.00% | 68.75% | 16.25% |
| Elastic (med) | 81.25% | 16.25% | 0.00% | 3.75% |
| Salt and pepper | 95.00% | 69.68% | 93.75% | 0.00% |

Table 3: Pairwise comparison matrix, noise distortion (UCF-101). Results are aggregated from 20 participants.

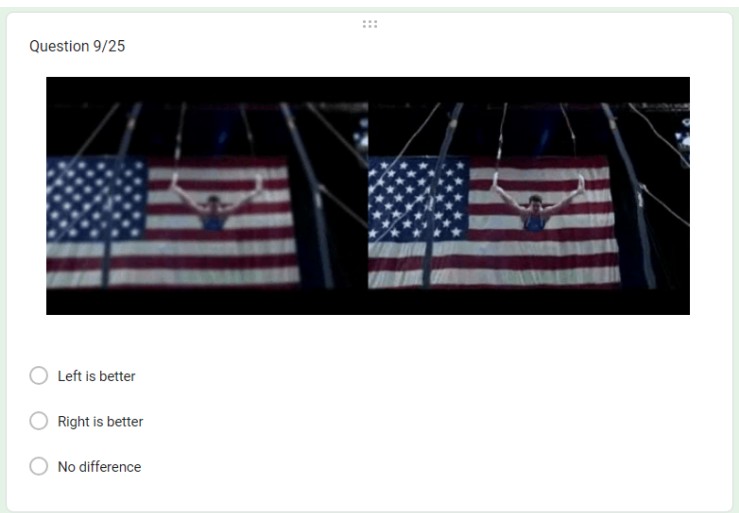

Figure 20: An example question in the human evaluation, noise distortion survey on the UCF-101 dataset.

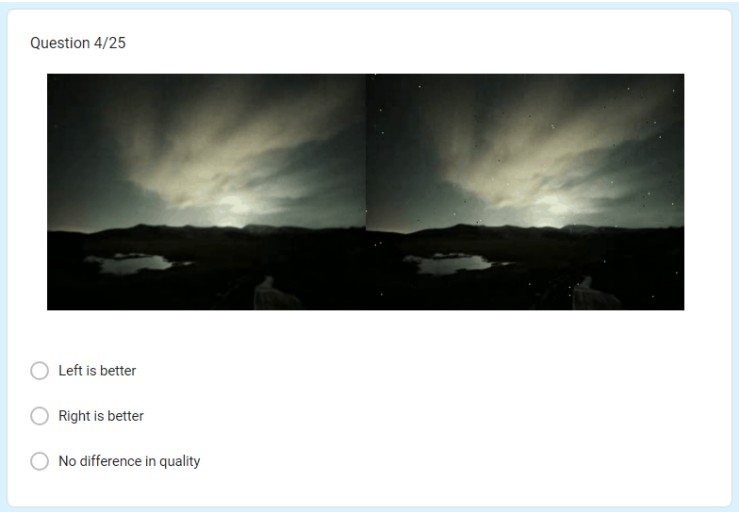

Figure 21: An example question in the human evaluation, noise distortion survey on the Sky Scene dataset.

| Noise Type | Blur (high) | Blur (medium) | Elastic (medium) | Salt and pepper |
|---|---|---|---|---|
| Blur (high) | 0.00% | 7.63% | 33.23% | 37.50% |
| Blur (med) | 68.23% | 0.00% | 41.25% | 48.75% |
| Elastic (med) | 56.78% | 42.50% | 0.00% | 41.25% |
| Salt and pepper | 53.75% | 46.25% | 51.25% | 0.00% |

Table 4: Pairwise comparison matrix, noise distortion (Sky Scene). Results are aggregated from 20 participants.

### F.4 Previous Human Studies for Image and Video Metrics

**Video Distributions - FVD**: To evaluate the alignment of FVD with human perception, Unterthiner et al. (2019) trained many variations of four video prediction models—CDNA, SV2P, SVP-FP, and SAVP—on the BAIR robot pushing dataset. For evaluation, they selected 256 video sequences and

| Rank | Method | Feature Space | UCF-101 (%) | Sky Scene (%) |
|------|--------|---------------|-------------|---------------|
| 1 | Energy | V-JEPA$_{PT}$ | 83.26 | 84.06 |
| 2 | MMD$_{RBF}$ | V-JEPA$_{PT}$ | 83.02 | 84.07 |
| 3 | MMD$_{LAP}$ | V-JEPA$_{PT}$+AE | 83.02 | 85.60 |
| 4 | FD | V-JEPA$_{PT}$ | 82.31 | 86.82 |
| 5 | MMD$_{POLY}$ | V-JEPA$_{PT}$ | 81.82 | 81.07 |
| 6 | Energy | V-JEPA$_{PT}$+AE | 80.88 | 83.17 |
| 7 | FD | V-JEPA$_{PT}$+AE | 79.33 | 84.13 |
| **8** | **MMD$_{POLY}$** | **V-JEPA$_{SSv2}$** | **78.96** | **78.39** |
| 9 | MMD$_{LAP}$ | V-JEPA$_{PT}$ | 78.85 | 81.34 |
| 10 | MMD$_{POLY}$ | V-JEPA$_{PT}$+AE | 78.78 | 82.33 |
| 11 | MMD$_{RBF}$ | V-JEPA$_{SSv2}$ | 75.51 | 79.10 |
| 12 | MMD$_{LAP}$ | V-JEPA$_{SSv2}$ | 75.49 | 79.98 |
| 13 | Energy | V-JEPA$_{SSv2}$ | 75.42 | 79.85 |
| 14 | MMD$_{RBF}$ | V-JEPA$_{PT}$+AE | 75.32 | 94.67 |
| 15 | FD | V-JEPA$_{SSv2}$ | 74.63 | 86.32 |
| 16 | MMD$_{RBF}$ | I3D+AE | 70.32 | 75.43 |
| 17 | MMD$_{RBF}$ | V-JEPA$_{SSv2}$+AE | 63.78 | 84.03 |
| 18 | FD | V-JEPA$_{SSv2}$+AE | 61.15 | 84.12 |
| 19 | Energy | V-JEPA$_{SSv2}$+AE | 59.12 | 79.09 |
| 20 | MMD$_{POLY}$ | V-JEPA$_{SSv2}$+AE | 58.47 | 78.60 |
| 21 | MMD$_{LAP}$ | V-JEPA$_{SSv2}$+AE | 57.93 | 80.34 |
| 22 | FD | VideoMAE$_{SSv2}$ | 55.72 | 79.08 |
| 23 | Energy | VideoMAE$_{SSv2}$ | 51.76 | 74.15 |
| 24 | MMD$_{RBF}$ | I3D | 50.93 | 68.07 |
| 25 | MMD$_{LAP}$ | VideoMAE$_{SSv2}$ | 50.16 | 72.76 |
| 26 | MMD$_{POLY}$ | VideoMAE$_{SSv2}$+AE | 49.76 | 75.54 |
| 27 | MMD$_{POLY}$ | VideoMAE$_{SSv2}$ | 48.34 | 75.35 |
| 28 | Energy | VideoMAE$_{SSv2}$+AE | 46.56 | 72.09 |
| 29 | MMD$_{RBF}$ | VideoMAE$_{SSv2}$ | 46.36 | 73.47 |
| 30 | MMD$_{POLY}$ | VideoMAE$_{PT}$ | 46.14 | 62.87 |
| 31 | MMD$_{RBF}$ | VideoMAE$_{PT}$ | 45.98 | 62.29 |
| 32 | MMD$_{LAP}$ | VideoMAE$_{SSv2}$+AE | 42.85 | 69.20 |
| 33 | FD | VideoMAE$_{SSv2}$+AE | 42.38 | 72.42 |
| 34 | Energy | VideoMAE$_{PT}$ | 40.03 | 60.73 |
| 35 | MMD$_{RBF}$ | VideoMAE$_{SSv2}$+AE | 37.90 | 73.47 |
| 36 | Energy | VideoMAE$_{PT}$+AE | 37.64 | 58.12 |
| 37 | MMD$_{LAP}$ | VideoMAE$_{PT}$+AE | 37.47 | 57.99 |
| 38 | MMD$_{POLY}$ | VideoMAE$_{PT}$+AE | 36.88 | 58.06 |
| 39 | FD | VideoMAE$_{PT}$ | 35.36 | 61.94 |
| 40 | MMD$_{RBF}$ | VideoMAE$_{PT}$+AE | 32.44 | 57.04 |
| 41 | FD | VideoMAE$_{PT}$+AE | 32.01 | 57.42 |
| 42 | FD | I3D+AE | 31.66 | 57.54 |
| **43** | **FD** | **I3D** | **31.32** | **58.07** |
| 44 | MMD$_{LAP}$ | VideoMAE$_{PT}$ | 29.90 | 57.93 |
| 45 | MMD$_{POLY}$ | I3D+AE | 25.97 | 67.25 |
| 46 | MMD$_{LAP}$ | I3D | 25.55 | 54.57 |
| 47 | Energy | I3D+AE | 25.52 | 48.07 |
| 48 | MMD$_{LAP}$ | I3D+AE | 25.17 | 57.30 |
| 49 | Energy | I3D | 23.74 | 47.25 |
| 50 | MMD$_{POLY}$ | I3D | 22.38 | 67.17 |

Table 5: Human Evaluation: Cosine Similarity Results (Ranked on UCF-101). VJEPA$_{SSv2}$+MMD$_{POLY}$ and FVD (I3D+FD) results are highlighted.

computed SSIM, PSNR, FVD, and KVD scores for each model. Both FVD and KVD metrics were calculated in the I3D feature space to assess perceptual similarity and distributional alignment. Up to 3 humans evaluated 3 videos from each model in one metric spread, one metric equal comparisons, where both comparison types involved 10 models. They reported that FVD was most aligned with humans, while KVD was similar but slightly worse.

**Image Distributions - CMMD**: In a study on the alignment of FID and CMMD (CLIP + MMD) with human evaluation, Jayasumana et al. (2024) ran a survey using two Muse models trained on the WebLI dataset, with one model (Model A) purposely trained for longer than the other (Model B). Side-by-side evaluations were performed where human raters were presented with two images, having to select which looked more preferable. All image pairs were rated by 3 independent raters, with 1633 prompts used in total. They reported that human raters and CMMD preferred Model A, while FID preferred model B.

**T2V Alignment Quality - T2VQA**: Kou et al. (2024) constructed a 10,000 sample dataset of AI-generated videos from 10 Text-to-Video (T2V) AI models. To determine the Mean Opinion Score (MOS) for each model, 27 subjects rated its perceptual quality based on two main criteria: text-video alignment - how well the video content matches the text description - and video fidelity - covering distortion, saturation, motion consistency, and content rationality. Subjects used a slider from 0 to 100 to assign their final score for each video. To amalgemate the results, the authors we conducted normalization to avoid inter-subject scoring differences.

# G GENERATIVE MODEL SPECIFICATIONS

⌂ Back to paper

## G.1 OPEN-SORA

For videos generated from the UCF-101 dataset, a single frame was provided as the image prompt with its corresponding video class label given as the text prompt. For videos generated from the Sky Scene dataset, a single frame was provided as the image prompt with "A sky timelapse" provided as the text prompt.

Videos were generated at an output resolution of 240p with an aspect ratio of 3:4. The '4s' preset for the number of frames was used which translates to 102 frames at 24 fps. These videos were converted to 16 frames at 7 fps for both the metric calculations. We used the open-source implementation loaded with the hpcai-tech/OpenSora-VAE-v1.2 checkpoint for inference.

## G.2 CTRL-V

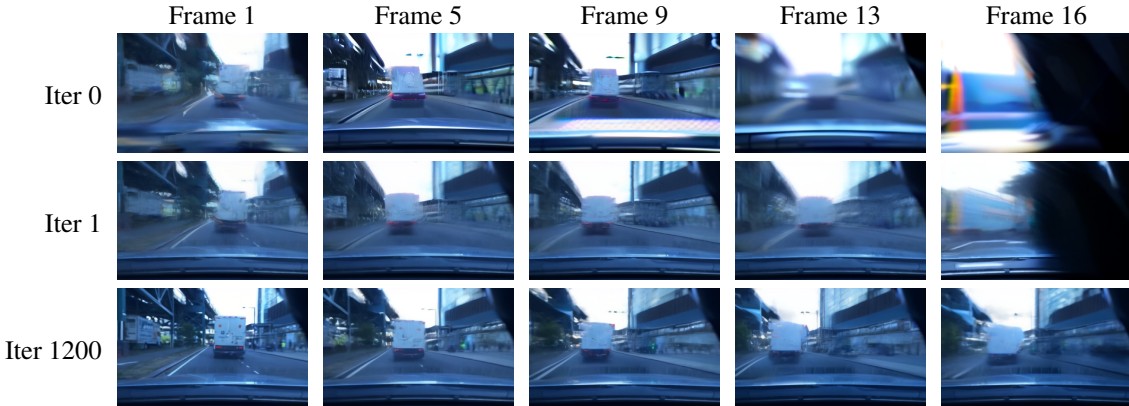

Figure 22: Training progression of Ctrl-v SVD generation at iteration 0, 1 and 1200.

⌂ Back to paper

We use the original code and model configuration from Ctrl-V (Luo et al., 2024) in this study, and their Stable Video Diffusion's model backbone is HuggingFace's `stabilityai/stable-video-diffusion-img2vid-xt` model. Figure 22 contains a demo of samples generated from the Ctrl-v's SVD checkpoints.

## H   A STUDY OF THE FEATURE SPACE WITH AI-GENERATED CONTENT

⌂ Back to paper

In this section, we demonstrate compelling results showing that JEDi is a superior metric compared to FVD when applied to AIGC.

To interpret these results, we need to agree on a key principle:

- If video sets A and B originate from the same distribution, their features—when extracted using any feature extractor—should remain in the same distribution. This is because a consistent extractor should not introduce artificial distinctions between sets that inherently belong to the same distribution.

- On the other hand, if video sets A and B come from different distributions, a robust feature extractor designed for evaluating distributional differences would highlight these distinctions. A good extractor would encode the unique characteristics of each set, creating clearly separable distributions in the feature space. This separation is critical for understanding the underlying differences and verifying that the extractor is effectively capturing meaningful distributional variations rather than noise or irrelevant features.

Validating a feature extractor's ability to consistently distinguish different distributions confirms its usefulness for evaluating distributional differences. Conversely, an extractor that fails to separate distinct distributions would be less effective in this context.

By this framework, we show that V-JEPA extracted features reveal clear distributional distinctions between 3 AIGC video sets and 1 real video dataset, whereas I3D features do not exhibit such distinct clustering.

Specifically, we utilize the generated samples from the TIP-I2V (Wang & Yang, 2024) generated by the Pika (Pika-AI, 2024), SVD (Blattmann et al., 2023), and Open-SORA (Zheng et al., 2024) models, as well as the real video samples from the WebVid10M dataset (Bain et al., 2021), and map them into I3D and V-JEPA feature spaces. Then, we employ LDA transformation to project these features into a 3D space, with the model label serving as the conditioning factor. The 3D slices of the feature space are provided in Figure 23. These illustrations show that JEPA-extracted features reveal clear distributional distinctions between 3 AIGC video sets and 1 real video dataset, whereas I3D features do not exhibit such distinct clustering.

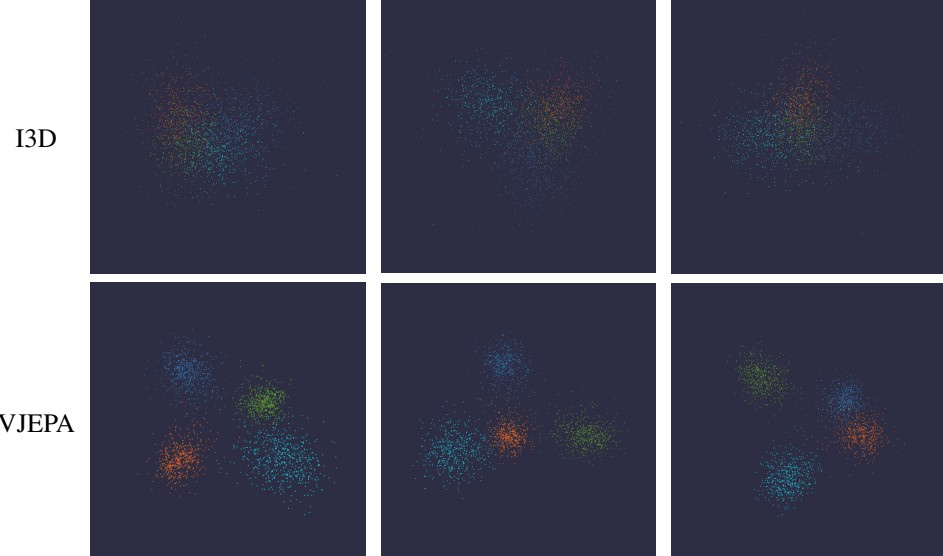

Figure 23: The 3D slices illustrate different viewpoints of the feature spaces, where the blue points represent **Pika** samples, the teal points represent **SVD** samples, the green points indicate **Open-SORA** samples, and the orange points indicate **WebVid10M** samples.

# I    ANALYSIS OF LONG-TERM VIDEOS

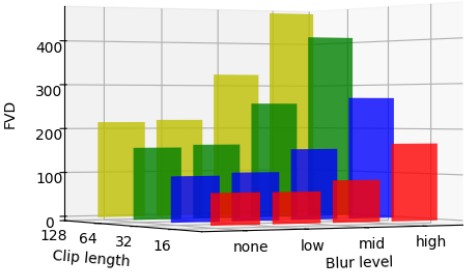 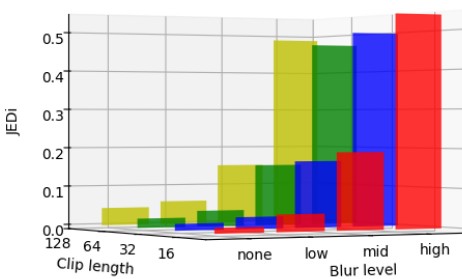

Figure 24: This figure illustrates the evaluated distances between the original training set and test sets with varying levels of blur distortion across different clip lengths. The bars are color-coded to represent different video clip lengths, while separate bars within each color group denote varying levels of distortion. The bar height indicates the measured distance. Notably, FVD exhibits a monotonic increase in measured distance with clip length within each distortion group. In contrast, JEDi yields constant measured distances within each distortion group, irrespective of clip length.

We conducted a study to assess the FVD and JEDi metrics across various clip lengths under different levels of blur distortion. The blur distortion settings followed those depicted in Table 1, with a uniform application of distortion across each clip. The results in Figure 24 show that for FVD, the evaluated distances increase as clip length grows for the same blur level. In contrast, JEDi distances remain consistent regardless of clip length.

This demonstrates that the JEDi metric is sensitive solely to the amount of distortion applied to the clips. When the same level of distortion is applied, JEDi evaluates the distance consistently, regardless of the clip length. In contrast, FVD is influenced by clip length, with evaluated distributional distances increasing as the clip becomes longer.

# J    LIMITATIONS

## J.1    JEDI: PERFORMANCE-EFFICIENCY TRADEOFFS

While we performed extensive study of different feature space with different distance metrics on different datasets, there are more datasets to be tested on and more types of noise distortions can be investigated. While we choose JEDi (MMD$_{POLY}$+V-JEPA$_{SSv2}$) as our main proposed method, other choices (see Table 5 for the list) such as the Energy distance with V-JEPA$_{PT}$ has slightly higher alignment with human evaluation. It is a decision we made considering the large gain in sample efficiency from JEDi.

## J.2    AVENUES FOR FUTURE RESEARCH

Similar to FVD, JEDi is a distribution-based evaluation metric that measures the discrepancy between the generated and ground-truth distributions in feature space. While our study focuses on distribution evaluation, our findings have implications for assessing the quality of individual video samples.

In prior literature on single video quality blind assessment, the Natural Video Statistics (NVS) model, proposed by Saad & Bovik, employed a multi-faceted approach, combining various models and techniques to extract features relevant to different aspects of video quality. This included hand-crafted temporal motion features, derived from differences between consecutive frames, and spatial characteristics captured using NIQE (Mittal et al., 2013) features. Our approach, on the other hand, integrates spatial and temporal aspects into a single, unified framework: the V-JEPA feature space.

The V-JEPA feature space is particularly well-suited for evaluating video quality, as it is sensitive to both spatial and temporal distortions, enabling a comprehensive assessment. This characteristic

presents a promising application: utilizing a large-scale, diverse, and high-quality video dataset mapped to the V-JEPA feature space to establish a pre-defined guideline distribution.

The quality of individual videos can be quantitatively assessed by mapping them to the V-JEPA feature space and computing their distance or likelihood relative to the guideline distribution, which serves as a robust benchmark for high-quality videos. Possible evaluation methods include computing the Mahalanobis Distance (Dodge, 2008) or the likelihood of the sample with respect to the distribution.

