# OpenReview forum: "Beyond FVD: An Enhanced Evaluation Metrics for Video Generation Distribution Quality"
_ICLR.cc/2025/Conference — ICLR 2025 Poster_

### Official Review · Reviewer_uZfY · 2024-10-16

**Soundness:** 3
**Presentation:** 3
**Contribution:** 3
**Rating:** 6
**Confidence:** 3

**Summary:**

The paper makes a strong empirical case for the JEDi metric, which is based on features derived from a Joint Embedding Predictive Architecture, measured using Maximum Mean Discrepancy with polynomial kernel, as a superior alternative to FVD in evaluating video generation models.

**Strengths:**

1. Interesting metric. The paper introduces the JEDi , a new metric for evaluating video generation quality, addressing several limitations of the existing FVD. JEDi is shown to improve alignment with human evaluations by 34%, which is a significant improvement.

2. Clear Identification of FVD’s Limitations. The authors thoroughly analyze the shortcomings of FVD, including:
    a.Non-Gaussianity in feature space.
    b. Insensitivity to temporal distortions.
    c.The impractically large sample sizes required for reliable metric estimation.

These are important challenges in the field of video generation, and addressing them demonstrates the paper's relevance.

3. Good Efficiency. One of the key claims is that JEDi requires only 16% of the samples compared to FVD to reach its steady value, which could make it a more practical metric for real-world applications.

**Weaknesses:**

First I want to declare that I am not very familiar with this area and I am going to summarize some weaknesses that matter to me.
1. Missing Analysis on Long-Term Temporal Distortions. Although the paper mentions the insensitivity of FVD to temporal distortions, the experiments primarily focus on shorter-term consistency and aesthetics. It would be beneficial to see deeper analysis on how JEDi handles long-term temporal coherence in video.

2. Issues About Human Evaluation.  Are these human subjects trained and qualified to evaluate video quality? Do the randomly selected videos adequately represent the entire set, and is there sufficient diversity? Has there been consideration of increasing the number of evaluation videos and including more subjects to ensure the reliability of the conclusions?

3.  In 5.1 section, how is the degree of blur determined, and are there any objective metrics for it? For the detection and assessment of noise and blur, many no-reference quality evaluation algorithms can perform this task. It is recommended to include some quality evaluation metrics for comparison, such as NIQE and Q-Align.

**Questions:**

See the weaknesses.

---

> ### Author Response · Authors · 2024-11-20
>
> Dear Reviewer,
>
> Thank you for your thorough review and insightful feedback.
>
> 1) Regarding **long-term video evaluation**, we are currently conducting experiments in this area. The results of this analysis will be incorporated into the next revised version of our manuscript.
> 2) Regarding the comment on **human evaluation subjects**, the participants include a diverse group of individuals, such as extended family members, friends, and fellow researchers. Since most media content is consumed by the average consumer, involving a randomly selected group of individuals is appropriate. Notably, many previous studies evaluating generative models have similarly relied on untrained participants for evaluating the generated content.
>
>     Additionally, we conducted a new study on the T2VQA-DB[1], a database containing recent Text-to-Video model generations along with human mean opinion scores (MOS). We use these MOS to evaluate the performance of FVD and JEDi. Both FVD and JEDi's results align with the MOS scores, with JEDi demonstrating significantly greater sample efficiency compared to FVD on this database (*Section 5.5, Pages 9-10*).
>
> 3) Regarding the comment on **the degree of blur distortion introduced in our experiments**, we have included the specific blur parameters in our revised manuscript.
>
>     Regarding the suggestion to **use metrics like NIQE and Q-Align for assessment**, we note that these metrics are designed for image assessment. However, our blur experiment introduces temporal randomness by sampling blur levels along the temporal dimension. Specifically, the sigma parameter controls the degree of blur applied to each frame, and we define a range for sigma from which the blur level for each frame is randomly sampled. The size of this sigma range correlates with the blur intensity: smaller ranges correspond to low temporal blur, while larger ranges correspond to high temporal blur. This approach introduces significant temporal inconsistency in the video, with abrupt transitions between clear and blurred frames. Since image-based metrics cannot capture these temporal inconsistencies, we utilize video-based networks (such as I3D, VideoMAE, and V-JEPA) to extract features and quantify these temporal distortions. Details of these experimental settings are revised in *Table 1 on page 7*. Additionally, it's important to note that the FVD backbone model, an I3D network trained for video action classification, primarily captures spatial features and may therefore struggle to detect temporal distortions.
>
> Additionally, in *Appendix G*, we present a new study analyzing samples generated by state-of-the-art video generation models within the I3D and V-JEPA feature spaces, using samples obtained from [2]. The findings from this experiment demonstrate that the V-JEPA feature space employed in JEDi provides a more efficient framework for evaluating video distribution.
>
> *[1] Kou, T., Liu, X., Zhang, Z., Li, C., Wu, H., Min, X., Zhai, G., & Liu, N. (2024). Subjective-Aligned Dataset and Metric for Text-to-Video Quality Assessment. ArXiv, abs/2403.11956.*
>
> *[2] Wang, W., & Yang, Y. (2024). TIP-I2V: A Million-Scale Real Text and Image Prompt Dataset for Image-to-Video Generation.*

---

> ### Author Response · Authors · 2024-11-22
> **Follow-up**
>
> Dear Reviewer,
>
> We have incorporated the **long-term video study** into our manuscript (*Section I, page 34*). The study concludes that the JEDi metric is sensitive only to the amount of distortion applied to the clips, maintaining consistent evaluations regardless of clip length. In contrast, FVD is influenced by clip length, with evaluated distributional distances increasing as the clip lengthens under the same level of distortion.
>
> Additionally, we have included **demographic information about the raters** for our human study (*Section F.3, Pg 29*).
>
> We hope these responses have addressed all of your concerns. We are also happy to discuss any remaining issues that may still be unclear. If we have clarified any confusion, we kindly ask that you consider revising your scores for our work.

---

> > ### Comment · Reviewer_uZfY · 2024-11-22
> > **Response to Authors**
> >
> > Thanks for the response. I would like to maintain my positive rating.

---

> > > ### Author Response · Authors · 2024-11-27
> > > **Follow-up II (New Updates)**
> > >
> > > Hi,
> > >
> > > Thank you so much for your positive feedback and time. We have made further improvements to our paper since your last response. Specifically,
> > > - We’ve added a new section based on the suggestions from Reviewer `goxo` and `NmJ7` (*Appendix F.4, pg 30-32*) that provides an overview of the human studies conducted for FVD, CMMD and T2VQA-DB.
> > > - We’ve recomputed all the figures in the paper and enhanced their readability (suggested by Reviewer `NmJ7`).
> > >
> > > We’d like to invite you to take a look at these new updates. We also want to mention that Reviewer `cX2Z` gave us a score of 3 (confidence 4) while having a review focusing on `SreamDiffusion[3] based on SD-Turbo or LCM`, which is completely unrelated to our work. This reviewer also remains completely unresponsive even after the paper revision deadline. As such, if you find that the paper addressed your original concerns and is more than marginally above the acceptance threshold, we kindly ask that you consider raising your score.
> > >
> > > Thanks again for your help and consideration! We are always happy to answer more questions and discuss more ways we could improve the quality of this paper.

---

### Official Review · Reviewer_gxbR · 2024-10-22

**Soundness:** 3
**Presentation:** 3
**Contribution:** 3
**Rating:** 8
**Confidence:** 4

**Summary:**

This paper provides critical analysis about FVD, which is the commonly used quality evaluation metric for generated videos. Three major limitations of FVD are discussed: 1) over-reliance on Gaussianity assumptions, 2) high-dimensional feature spaces, and 3) low sample efficiency. Furthermore, this paper proposes a new metric named JEDi, which computes Maximum Mean Discrepancy (MMD) in a V-JEPA feature space. Extensive experiments are conducted and the results demonstrate its robustness and effectiveness in terms of distortion-sensitivity, high sample efficiency, and high correlation with human ratings.

**Strengths:**

1.	The limitations of FVD are systematically identified and analyzed, especially regarding the non-Gaussianity of the I3D feature space and the low sample efficiency.
2.	Findings and insights derived from comprehensive comparison experiments are interesting and convincing, such as the importance comparison of feature space and distance metrics, and the distortion-aware feature space selection.
3.	A well-justified metric name JEDi is proposed that addresses the main issues in FVD. The methodology is clearly explained, and the effectiveness of JEDi is well verified through extensive experiments.

**Weaknesses:**

1. Due to the limitations of the small-scale human evaluation, it would be better to evaluate the quality evaluation performance on large-scale video quality assessment datasets of generated content. In the human evaluation, only comparison ratings as the human preference are provided, lacking the overall quality ratings which can be obtained by single stimulus. Since JEDi is targeting quality evaluation for generated videos where the quality is practically measured without reference, i.e., no-reference quality assessment, it would be more comprehensive to evaluate JEDi on large-scale open-source video quality databases of AI-generated content, such as T2VQA-DB [1] and GAIA [2].
2. Lacking of reviewing and analyzing relevant literatures. Similar to FID ant its variants, NIQE [3] evaluates the quality by measuring distribution distance in the feature space. Targeting measure quality of non-generated images, NIQE extracts neurostatistical features, fit the features with Multivariate Gaussian Model, and compute distribution distance by Mahalanobis Distance. It would be very insightful to include the Mahalanobis Distance into the discussion of the effective distance metrics, and to analyze the difference of algorithm design among FID/FVD, NIQE, and JEDi. Quantitive experiments are expected to provide holistic analysis.

[1] GAIA: Rethinking Action Quality Assessment for AI-Generated Videos, arXiv 2024
[2] Subjective-Aligned Dateset and Metric for Text-to-Video Quality Assessment, arXiv 2024
[3] Making a “completely blind” image quality analyzer[J]. IEEE Signal processing letters, 2012

**Questions:**

Can the monotonicity with training iterations in Fig. 5 imply the accuracy as well? Fig.5 only demonstrate the better monotonicity of JEDi during training over that of FVD and other metrics, yet it can not tell the superiority of quality evaluation accuracy of JEDi. Refer to the commonly used index, SROCC, PLCC, and RMSE, for measuring the performance of quality assessment metrics.

---

> ### Author Response · Authors · 2024-11-15
>
> Dear Reviewer,
>
> Thank you for your thorough review and insightful feedback.
>
> JEDI and FVD are generative distribution assessment metrics that rely on reference datasets for evaluations (cannot be used for no-reference quality assessment). We appreciate your recommendation to evaluate our metric using AI-generated conten, and we conducted experiments on the T2VQA database, comparing the JEDi and FVD distances.
>
> Firstly, the T2VQA database has the following characteristics:
> - Each video is associated with a single Mean Opinion Score (MOS) aggregating ratings for text-video alignment and video fidelity, obtained through a 0-100 slider (Section 3.2 of [1]).
> - Human-evaluated scores exhibit significant variance, clustering around 50 with a standard deviation of 16.6 (Figure 3 and Section 3.2 of [1]).
> - The database contains generations from 10 different AI models. Many models yield similar MOS scores, resulting in an ambiguous ranking of the models if used directly (Figure 3 of [1]).
> - T2VQA-DB lacks a "ground-truth" video set. To address this, we identified corresponding videos from WebVid10M using the same prompts and utilized them as our ground-truth distribution for evaluation.
>
> We selected generations from three models (#2, 7, and 9) that exhibit a clear gap in video quality, with average rankings of 10 (worst), 5, and 1 (best), respectively, based on MOS scores. Both FVD and JEDi scores ranked these models consistently, with model #2 as the worst and model #9 as the best. Our evaluation results are
> |                                 | Model #2    | Model #7    | Model #9    |
> |---------------------------------|-------------|-------------|-------------|
> | Ranks according to MOS          | Worst          | Median           | Best           |
> | JEDi (#. conv samples) | 4.427 (100) | 2.164 (200) | 0.722 (300) |
> | FVD (#. conv samples)  | 884.1 (600) | 445.6 (800) | 279.9 (900) |
>
>  Importantly, JEDi achieved convergence with significantly fewer samples, compared to FVD: the number of samples to converge for JEDi are 100, 200 and 300; the number of samples to converge for FVD are 600, 800 and 900.
>
> We initially planned to evaluate JEDi and FVD on the LGVQ and GAIA datasets. However, the LGVQ dataset is not yet available, and GAIA lacks a ground-truth set, with its scores also reflecting action-tag alignment. We are currently exploring alternative AIGC datasets that offer decoupled visual quality scores from human raters alongside a ground-truth set. We will keep you updated on our progress.
>
> Regarding your comment on NIQE evaluator:
> We acknowledge NIQE's effectiveness in image quality assessment, but highlight its limitations for video data. Specifically:
> - Image evaluators, like NIQE, overlook temporal consistency crucial for video evaluation.
> - NIQE assesses single-image quality, whereas our work focuses on evaluating generation distribution quality, which examines a model's generalization capability.
> - NIQE's reference-free approach differs from our metric, which measures distance from the ground-truth distribution.
>
> Addressing your concern about the monotonicity assumption in Figure 5, we provide visualizations of generation results at various training checkpoints in Figure 22 – the video version of which is uploaded here: [link](https://accio-muggle.github.io/anonymous-results.github.io/). As shown, the generated videos are visibly higher quality as the number of iterations increases.
>
> *[1] Kou, T., Liu, X., Zhang, Z., Li, C., Wu, H., Min, X., Zhai, G., & Liu, N. (2024). Subjective-Aligned Dataset and Metric for Text-to-Video Quality Assessment. ArXiv, abs/2403.11956.*

---

> > ### Author Response · Authors · 2024-11-21
> > **Follow-up**
> >
> > Dear Reviewer,
> >
> > Thank you again for your valuable feedback. We have revised our manuscript accordingly:
> > 1) We have incorporated new experiment results using the T2VQA database [2] into *Section 5.5 (Pg 9-10)*, as previously mentioned in our response to the feedback concerning **the assessment of AIGC materials**.
> > 2) We conducted an additional experiment using the TIP-I2V dataset [1], comprising samples generated by current state-of-the-art video generation models. Analyzing these video samples in the I3D and V-JEPA feature spaces provides valuable insights into our metric's performance. The details of this experiment are provided in *Appendix G (Pg 32)*, with a brief discussion included in *Section 5.5 (Pg 9-10)* of the main text.
> >
> > *[1] Wang, W., & Yang, Y. (2024). TIP-I2V: A Million-Scale Real Text and Image Prompt Dataset for Image-to-Video Generation.*
> >
> > *[2] Kou, T., Liu, X., Zhang, Z., Li, C., Wu, H., Min, X., Zhai, G., & Liu, N. (2024). Subjective-Aligned Dataset and Metric for Text-to-Video Quality Assessment. ArXiv, abs/2403.11956.*
> >
> >
> > We hope these responses have addressed all of your concerns. We are also happy to discuss any remaining issues that may still be unclear. If we have clarified any confusion, we kindly ask that you consider revising your scores for our work.

---

> > ### Comment · Reviewer_gxbR · 2024-11-22
> > **Feedback to authors**
> >
> > Thanks for the feedback. There are still some issues that may warrant further discussion:
> > 1. Reference or No-reference? As far as I understand, the Image/Video Quality Assessment (IQA/VQA) of AI-generated content should generally fall under the no-reference IQA/VQA category rather than the full-reference category, since AI-generated content typically lacks a corresponding reference counterpart. Metrics such as FID and FVD are no-reference metrics that measure the distribution distance between generated content and a pre-defined dataset. Note that the pre-defined dataset is not technically a reference for the specific generated content. Instead, it serves as an off-the-shelf prior distribution for any generated content, making FID and FVD inherently no-reference metrics. For technically full-reference video quality assessment, databases like LIVE-VQA and MCL-V are relevant examples.
> > 2. Concerns on NIQE. In my opinion, there are common design insights between NIQE and FID/FVD, which is feature extraction-->distribution fitting-->distribution distance computation. Similar to FID/FVD, where a pre-defined dataset serves as the prior distribution reference, NIQE adopts a set of high-quality images as the pre-defined content, computes natural scene statistics-based features, fits features with Multivariate Gaussian Model (MVG), computes distributions distance between pre-defined content and the target content by Mahalanobis Distance. In your appendix, several distance metrics are discussed, and I believe that Mahalanobis Distance could be included in discussion or even evaluated in FVD/JEDi, if possible.

---

> > > ### Author Response · Authors · 2024-11-22
> > > **Question about Mahalanobis Distance**
> > >
> > > Dear Reviewer,
> > >
> > > Thank you for your valuable feedback and suggestions. Before we respond to your question, could you kindly clarify something for us:
> > >
> > > To our understanding, **the Mahalanobis distance measures the distance between a point and a distribution**, typically used to assess whether a point belongs to a given distribution. However, the problem we are addressing in this paper is testing whether the generated distribution differs from the ground-truth distribution and, if so, by how much. Therefore, the statistical metrics we have tested in this work, such as Fréchet distance or energy statistics, are **designed to measure the distance between two distributions** (or, in simpler terms, the distance between two point clouds).
> > >
> > > The Mahalanobis distance, or similar metrics, are generally used to measure the likelihood of a single sample. As you mentioned, NIQE fits the extracted features into a multivariate Gaussian (MVG) model and computes the likelihood of an image based on natural scene statistics. However, this metric is not suitable for measuring distributional distances.
> > >
> > > Please correct us if we are mistaken. As far as we know, the Mahalanobis distance between a vector $x$ and a distribution with mean vector $\mu$ and covariance matrix $\Sigma$ is defined as:
> > > $$D_M(x) = \sqrt{(x-\mu)^T\Sigma^{-1}(x-\mu)}$$
> > >
> > > If our generative model consistently produces the same video, where the video feature's mean equals $\mu$, the averaged Mahalanobis distance for such a generation set would always be 0. This highlights why distributional metrics, such as FVD, are essential for assessing the quality of the generated distribution.
> > >
> > > Please let us know if our understanding aligns with yours, and if there is anything we misunderstood, we would appreciate your clarification.
> > >
> > > Best regards.

---

> > > > ### Comment · Reviewer_gxbR · 2024-11-23
> > > > **Feedback on Mahalanobis Distance**
> > > >
> > > > Thank you for your clarification on Mahalanobis Distance. Since the Mahalanobis distance measures the distance between a point and a distribution instead of the distance between two distributions, there is no need to evaluate this distance in JEDi. Nevertheless, it is interesting to discuss the possibility of JEDi to evaluate a single generated video by calculating Mahalanobis Distance, which is also concerned by Reviewer goxo.

---

> ### Author Response · Authors · 2024-11-23
> **Comment regarding the references for FVD and JEDi metrics**
>
> Dear Reviewer,
>
> Thank you for your feedback regarding **the references for FVD and JEDi metrics**. We would like to clarify that these metrics **indeed require reference distributions**.
>
> To ensure clarity, we would like to define the following terminologies:
> - **Generic video distribution (also referred to as a pre-defined or off-the-shelf prior video distribution in your reviews):** A video distribution that serves as a broad summary of all types of videos, encompassing samples from the entire real video space.
> - **Target distribution (also known as the ground-truth distribution):** The desired distribution that we aim for our generative model to replicate (e.g. to evaluate a generative model's driving scene generation capability, the target distribution should contain exclusively driving videos and should not be a generic video distribution.)
> - **Distributional metrics:** Metrics that measure the distance between the generated distribution and the target distribution.
>
> We would like to address two key points:
> 1) FVD is **a specific type of metric within the VQA family**, designed to measure distributional distances between two sets of features using statistical tools (i.e. Fréchet distance and optimal transport) tailored for distribution-to-distribution comparisons.
> 2) For metrics like FVD, **accurate evaluations rely on precisely defining the target distribution**. Using a generic video dataset as a pre-defined target distribution can be inappropriate, particularly when dealing with specialized domains. For instance, autonomous vehicle (AV) videos occupy a unique subspace within the broader video space, with distinct mean and variance characteristics. Evaluating an AV video generation model against a generic video distribution would yield inaccurate results (*Figure 2 on Pg 4 visualizes the distribution of 11 video sets within the PCA-reduced feature space, revealing distinct clustering patterns for each video dataset*).
>
> To summarize, the proposed metric JEDi, similar to FVD, evaluates the distribution-to-distribution distance in feature space. This distributional property is crucial since non-distributional metrics solely focus on sample quality and overlook distribution bias. For example, let there be a generative model that is trained on a variety of driving videos but is biased towards generating one type of video (e.g., cars always turning left but never right): while non-distributional metrics would give these generations a high overall score if the video qualities are good, a distributional metric can detect the bias in the generated video distribution since it doesn’t match with the target distribution (*This concept is also discussed in our manuscript on lines 53–57, page 1-2, introduction section*).
>
> We hope our explanations have provided clarity on the concept of distributional metrics. If there are any remaining questions or concerns, we are more than happy to continue the discussion to ensure everything is fully addressed.

---

> > ### Comment · Reviewer_gxbR · 2024-11-23
> > **Feedback on reference metrics**
> >
> > Thank you for your clarification. I totally agree with you that these metrics require reference distributions to calculate the distribution distance between reference distribution and distribution of videos generated by an evaluated generative model. In my opinion, the same reference content (e.g., videos with high-quality motion) can serve as reference distribution for evaluating multiple generative models (e.g., evaluating if these generative models can generate videos with high-quality motion). In that case, when evaluating a new generative model, one can use these metrics as no-reference metrics with default reference distribution, which is similar to the widely-used case of NIQE as a no-reference metric to evaluate the naturalness of images (since the default reference content of NIQE is a set of high-quality natural images).

---

> ### Author Response · Authors · 2024-11-23
> **Follow-up III**
>
> Dear Reviewer,
>
> Thank you for your valuable feedback. Since our work centers on evaluating the quality of video distributions, we greatly appreciate your and Reviewer `goxo`'s suggestions to discuss our findings' implications for assessing individual videos.
>
> Thus, we have provided a comprehensive discussion in *Section J.2 (Pages 34-35)*, exploring the connections between our research and blind single video quality assessment. Specifically, we delve into the relationships with existing methods and related topics, including NIQE [2], Natural Video Statistics (NVS) [1], and Mahalanobis distance. We propose that the insights derived from the V-JEPA feature space have the potential to advance single-video quality metrics. Furthermore, we outline several preliminary experiments, such as utilizing Mahalanobis Distance to measure sample-distribution distances, which could lay the groundwork for future research.
>
> *[1] M. A. Saad and A. C. Bovik, "Blind quality assessment of videos using a model of natural scene statistics and motion coherency," 2012 Conference Record of the Forty Sixth Asilomar Conference on Signals, Systems and Computers (ASILOMAR), Pacific Grove, CA, USA, 2012, pp. 332-336, doi: 10.1109/ACSSC.2012.6489018.*
>
> *[2] A. Mittal, R. Soundararajan and A. C. Bovik, "Making a “Completely Blind” Image Quality Analyzer," in IEEE Signal Processing Letters, vol. 20, no. 3, pp. 209-212, March 2013, doi: 10.1109/LSP.2012.2227726.*
>
> We welcome any additional feedback you may have and hope that our revisions have adequately addressed your concerns.  If we have clarified any confusion, we kindly ask that you consider revising your scores for our work.

---

> ### Comment · Reviewer_gxbR · 2024-11-24
> **Final feedback to authors**
>
> Thanks for the authors' feedback. My concerns are all well addressed. I have updated my score to 8.

---

### Official Review · Reviewer_NmJ7 · 2024-10-23

**Soundness:** 2
**Presentation:** 2
**Contribution:** 3
**Rating:** 6
**Confidence:** 3

**Summary:**

This work analyzes the limitations of FVD and proposes the JEPA Embedding Distance based on features derived from a Joint Embedding Predictive Architecture. The authors commit to develop effective, robust metrics with small sampling size for guiding the current surge in video generation research. The experiments on open-source datasets show that compared to the widely used FVD metric, the proposed method requiring only 16% of the samples to reach its steady value, while increasing alignment with human evaluation by 34%, on average.

**Strengths:**

1. The authors note three key hurdles in video generation: (1) Data size, (2) Computational resources, and (3) Metric convergence speed, and commence by developing metrics with higher sample efficiency with fast convergence.
2. The authors investigate the correlation between metric with distortion level and training duration and gain valuable observations to further improve the FVD metric.

**Weaknesses:**

1. The content arrangement is not sufficiently appropriate. For example, Section 2.1 and 2.2 introduce a lot of consensus information about Inflated 3D ConvNet, video masked autoencoder, and the definition of Fr ́echet Distance, which should be moved to Appendix. In contrast, more related work about the evaluation metrics for video generation should be reviewed [1, 2].
2. The organization of Section 1 should be improved to enhance its readability. Line53: a video generation model …… produce ….. with diverse features? The two examples in Line53-57 are limited in scenarios. Line 59 - 60 only mentioned some full-reference video quality metrics,  however, lack of the introduction of no-reference video quality assessment (VQA) metrics ([1, 2]), which are more practical in real-world applications.
3. Fig. 3 is difficult to interpret without zooming in. Improving the resolution, providing clearer annotations or changing the scale could make the results more convincing.
4. What is the meaning of the items in red font in Table 1? Lack of explanation. And what is the meaning of blur (low, mid, high)? Are they mean the distortion Level? These should be noted in the Table caption.
5. The experiment is not comprehensive enough. Only two I2V models and three noise distortion types are included. I recommend that the authors add more image-to-video models for evaluation, both proprietary and open source, to support your findings. Since there are low-level distortion (blur, noise, artifacts) and high-level distortion (semantic, composition) in generated videos, it is necessary to evaluate the proposed JEDi in these scenarios.
6. The details of human evaluation in Section5.4 are vague. Since the normativity of the subjective study largely affect the reliability of these results. The content in App. E.3 is empty.
7. Line378: I3D and VideoMAEPT are not ideal feature spaces for building video quality metrics, as they do not capture blur distortion well. How about other distortions?
8. The equations in App. B should be better arranged to increase their readability.

[1] Ghildyal, Abhijay, et al. "Quality Prediction of AI Generated Images and Videos: Emerging Trends and Opportunities." *arXiv preprint arXiv:2410.08534* (2024).

[2] Min, Xiongkuo, et al. "Perceptual video quality assessment: A survey." *arXiv preprint arXiv:2402.03413* (2024).

**Questions:**

see weaknesses.

---

> ### Author Response · Authors · 2024-11-15
>
> Dear Reviewer,
>
> Thank you for your valuable feedback. We will revise our writing based on your suggestions.
>
> *For Points 1 & 2:*
> While many studies focus on single-video generation quality, our work centers on a different aspect: assessing generation distribution quality. To our knowledge, the only other work addressing this is FVD.
> Evaluating video distributions requires mapping the videos into feature spaces. For example, the I3D network used in FVD was trained on a human-action classification task; its features are biased toward spatial content (for recognizing actions) but are less suitable for evaluating temporal consistency. In contrast, VideoMAE and V-JEPA features, pretrained on self-supervised tasks, may be more appropriate for assessing overall video quality. Therefore, we believe defining the feature extractors and their feature space characteristics is important to facilitate understanding of distribution evaluation throughout the paper.
>
> *For Point 3:*
> We will adjust the scaling of the figures to improve readability.
>
> *For Point 4:*
> In the Table 1 experiment, the testing video dataset is subjected to noise distortions, including low blur (σ ∼ [0.05, 0.75]), medium blur (σ ∼ [0.1, 1.5]), and high blur (σ ∼ [0.01, 3]), where σ represents the per-frame blur intensity. A larger range indicates greater temporal inconsistency. We will update this information in the table caption.
>
> *For Points 5 & 6:*
> We will add new experimental results using the T2VQA database, which contains 10K generated videos from 10 AI models along with human evaluation scores. Key details of the database include:
> - Each video is associated with a single Mean Opinion Score (MOS) aggregating ratings for text-video alignment and video fidelity, obtained through a 0-100 slider (Section 3.2 of [1]).
> - Human-evaluated scores exhibit significant variance, clustering around 50 with a standard deviation of 16.6 (Figure 3 and Section 3.2 of [1]).
> - The database contains generations from 10 different AI models. Many models yield similar MOS scores, resulting in an ambiguous ranking of the models if used directly (Figure 3 of [1]).
> - T2VQA-DB lacks a "ground-truth" video set. To address this, we identified corresponding videos from WebVid10M using the same prompts and utilized them as our ground-truth distribution for evaluation.
>
> We selected generations from three models (#2, 7, and 9) that exhibit a clear gap in video quality, with average rankings of 10 (worst), 5, and 1 (best), respectively, based on MOS scores. Both FVD and JEDi scores ranked these models consistently, with model #2 as the worst and model #9 as the best. Our evaluation results are
> |                                 | Model #2    | Model #7    | Model #9    |
> |---------------------------------|-------------|-------------|-------------|
> | Ranks according to MOS          | Worst          | Median           | Best           |
> | JEDi (#. conv samples) | 4.427 (100) | 2.164 (200) | 0.722 (300) |
> | FVD (#. conv samples)  | 884.1 (600) | 445.6 (800) | 279.9 (900) |
>
>  Importantly, JEDi achieved convergence with significantly fewer samples, compared to FVD: the number of samples to converge for JEDi are 100, 200 and 300; the number of samples to converge for FVD are 600, 800 and 900.
>
> Additionally, we are exploring alternative AIGC datasets that provide decoupled visual quality scores from human raters and include a ground-truth set. We will keep you updated on our progress.
>
> Also, thank you for pointing out the formatting issue in Section E.3. Table 4 was intended to be under Section E.3 but was mistakenly shifted. We will correct this in the updated file.
>
> *For Point 7:*
> We have included plots of I3D and VideoMAE's feature sensitivities to various distortions in Figures 17 and 18. Figure 17 demonstrates that I3D features are highly sensitive to spatial distortion (salt & pepper noise), while Figure 18 shows that VideoMAE-PT features perceive medium elastic distortion as worse than high elastic distortion on the UCF dataset. Additionally, we included the correlation between evaluated feature distances under different distortions and human opinions in Table 4.
>
> *For Point 8:*
> We will revise the equations in Appendix B to improve readability.
>
> Thank you for your feedback. We appreciate your insights and will incorporate these changes in our updated submission.
>
> *[1] Kou, T., Liu, X., Zhang, Z., Li, C., Wu, H., Min, X., Zhai, G., & Liu, N. (2024). Subjective-Aligned Dataset and Metric for Text-to-Video Quality Assessment. ArXiv, abs/2403.11956.*

---

> ### Author Response · Authors · 2024-11-21
> **Follow-up**
>
> Dear Reviewer,
>
> Thank you again for your valuable feedback. We have revised our manuscript accordingly:
>
> 1) We have revised *Figure 3 (Pg 6)* to improve clarity.
> 2) In *Table 1 (Pg 7)*, red font highlights an inconsistency in FVD evaluation, where the distribution distance between*test and train* exceeds that between *test+low-blur and train*. We provide detailed experiment settings in the table caption to clarify any potential confusion.
> 3) We have incorporated new experiment results using the T2VQA database [2] into *Section 5.5 (Pg 9-10)*, as outlined in our prior response addressing feedback **Points 5 & 6**.
> 4) We conducted an additional experiment using the TIP-I2V dataset [1], comprising samples generated by current state-of-the-art video generation models. Analyzing these video samples in the I3D and V-JEPA feature spaces provides valuable insights into our metric's performance. The details of this experiment are provided in *Appendix G (Pg 32)*, with a brief discussion included in *Section 5.5 (Pg 9-10)* of the main text.
> 5) We have added a new section (*Appendix B (Pg 17-18)*) to discuss the computational overheads and distribution metric configurations of our work.
>
> *[1] Wang, W., & Yang, Y. (2024). TIP-I2V: A Million-Scale Real Text and Image Prompt Dataset for Image-to-Video Generation.*
>
> *[2] Kou, T., Liu, X., Zhang, Z., Li, C., Wu, H., Min, X., Zhai, G., & Liu, N. (2024). Subjective-Aligned Dataset and Metric for Text-to-Video Quality Assessment. ArXiv, abs/2403.11956.*
>
> We hope these responses have addressed all of your concerns. We are also happy to discuss any remaining issues that may still be unclear. If we have clarified your confusion, we kindly ask that you consider revising your scores for our work.

---

> > ### Comment · Reviewer_NmJ7 · 2024-11-21
> > **Feedback to authors**
> >
> > Overall, author's feedback addressed most of my concerns. I have updated my rating to 5. However, the manuscript could be further improved to enhance its readability (including the arrangment). The axis font in figure 2, 7, 8, 9, 10, 11, 12, 13, 14, 15, 16, 17, 18, 19, etc, is really too small and the content is hard to read.
> > Other questions:
> > 1. How many participants were recruited in the human evaluation? (same concern as Reviewer uZfY- weakness2). Roughly report the group of individuals may not convincing. A detailed age or sex distribution should be reported (as most IQA or VQA studies do).
> > 2. The application of this method should be disscussed further. Which assessment scenarios would be benefitted by this distribution-based method compared to quality assessment for single video.

---

> > > ### Author Response · Authors · 2024-11-23
> > > **Follow-up II**
> > >
> > > Dear Reviewer,
> > >
> > > Thank you for your follow-up feedback. As we continue to improve our manuscript, we would like to address the questions you raised:
> > >
> > > 1) We recruited 20 independent raters for each of the two surveys (*ref. Line 449, Pg 9*). Demographic information about the raters has been included in our appendix (*Appendix F.3, Pg 29*)
> > >
> > > 2) The primary advantage of distributional-based metrics lies in their ability to assess a generation model’s generalization capability, whereas non-distributional metrics focus solely on sample quality, thereby overlooking distributional bias.  For example, let there be a generative model that is trained on a variety of driving videos but is biased towards generating one type of video (e.g., cars always turning left but never right): while non-distributional metrics would give these generations a high overall score if the video qualities are good, a distributional metric can detect the bias in the generated video distribution since it doesn’t match with the target distribution (*This concept is also discussed in our manuscript on lines 53–57, page 1-2, introduction section*).
> > >
> > > We will continue to refine our manuscript. If you have any additional questions or concerns, please do not hesitate to contact us. Best Regards.

---

> > > ### Author Response · Authors · 2024-11-25
> > > **Follow-up III for Reviewer NmJ7**
> > >
> > > Dear Reviewer,
> > >
> > > Thank you for your feedback. We have **recomputed all the figures** you mentioned and incorporated them into the revised manuscript.
> > >
> > > We are happy to address any remaining concerns you may have. If you feel our manuscript has improved, we kindly request that you consider revisiting your evaluation. Best regards.

---

> > > > ### Comment · Reviewer_NmJ7 · 2024-11-25
> > > > **Update the rating**
> > > >
> > > > Thank you for your comprehensive clarification. My questions and concerns have been addressed satisfactorily. I thereby change my rating to 6.

---

> ### Comment · Reviewer_goxo · 2024-11-24
>
> Most of the previous concerns are addressed.
> Some surveys for image quality assessment as well as video quality assessment are suggested to be given in the paper, for better referring of the related topics.
> I have raised my score.

---

> ### Author Response · Authors · 2024-11-25
> **Response to Reviewer goxo**
>
> Thank you for suggesting incorporating a discussion of prior work's human studies. We have added a new survey section (*Appendix F.4, pg 30-32*), which provides an overview of the human studies conducted for **FVD (video distribution-based metric)** [1], **CMMD (image distributional-based metric)**[2], and **T2VQA-DB (AIGC video database)** [3].
>
> *[1] Unterthiner, T., Steenkiste, S.V., Kurach, K., Marinier, R., Michalski, M., & Gelly, S. (2018). Towards Accurate Generative Models of Video: A New Metric & Challenges. ArXiv, abs/1812.01717.*
>
> *[2] Jayasumana, S., Ramalingam, S., Veit, A., Glasner, D., Chakrabarti, A., & Kumar, S. (2023). Rethinking FID: Towards a Better Evaluation Metric for Image Generation. 2024 IEEE/CVF Conference on Computer Vision and Pattern Recognition (CVPR), 9307-9315.*
>
> *[3] Kou, T., Liu, X., Zhang, Z., Li, C., Wu, H., Min, X., Zhai, G., & Liu, N. (2024). Subjective-Aligned Dataset and Metric for Text-to-Video Quality Assessment. ArXiv, abs/2403.11956.*

---

### Official Review · Reviewer_goxo · 2024-11-03

**Soundness:** 3
**Presentation:** 3
**Contribution:** 3
**Rating:** 8
**Confidence:** 5

**Summary:**

This paper reveals that FVD falls short as a standalone metric for video generation evaluation, then analyzes a wide range of metrics and backbone architectures, and proposes JEDi, the JEPA Embedding Distance, based on features derived from a Joint Embedding Predictive Architecture, measured using Maximum Mean Discrepancy with polynomial kernel.

**Strengths:**

The proposed JEDi eliminates the need for parametric assumptions about the underlying video distribution, unlike FVD which relies on the Gaussianity assumption to make its metric feasible.
JEDi significantly reduces the number of samples needed to make an accurate estimate.
JEDi has better alignment with human evaluations compared to FVD.

**Weaknesses:**

1.	The tile of the paper is relatively too broad, as video generation quality as well as its evaluation cover many aspects, however JEDi can only cover an aspect of quality.
2.	The proposed JEDi is a distribution-based evaluation metric, which can not work for a single case, for example to evaluate the quality of a single generated video.
3.	The single video AIGC video quality assessment should also be discussed and compared, including the state-of-the-arts and its differences and similarities with the distribution metrics.
4.	The authors conduct a subjective evaluation which is a very small scale study. More evaluations on existing large-scale AIGC video quality databases should be conducted.
5.	More comparisons with the single AIGC video quality assessment metrics are suggested to be included.

**Questions:**

Following the above weaknesses, some questions are suggested to be answered.
1.	As we known, video generation quality covers many aspects, thus the authors are suggested to specify which aspect of quality this paper focuses on.
2.	The possibility of applications on single video quality evaluation should be discussed.
3.	The are many studies for single generated image or video quality assessment studies in the image/video quality assessment communities.
4.	In some open AIGC video quality assessment databases, both generated videos and human labels are given, for example the Text-to-Video Quality Assessment DataBase (T2VQA-DB), Large-scale Generated Vdeo Quality assessment (LGVQ) dataset, Generic AI-generated Action (GAIA) dataset. The alignment with human ratings on these databases are suggested to be tested.
5.	Following the above comment, the performance comparisons with these AIGC video quality models are suggested to be given.

---

> ### Author Response · Authors · 2024-11-15
>
> Dear Reviewer,
>
>
> Thank you for your insightful feedback. We appreciate your suggestion and acknowledge that our paper focuses specifically on evaluating video distribution generation quality, which is one aspect of broader video qualities. To clarify this scope, we plan to revise our title and abstract accordingly.
>
>
> Regarding the potential application of our findings to single video evaluation, this work highlights the advantages of evaluating video quality within the V-JEPA feature space over I3D and VideoMAE spaces. Based on this, we believe that the V-JEPA could serve as an effective extractor for single video quality evaluation by comparing the likelihood of a video against ground-truth video sets. Although this direction was not explored in the current paper, it offers a promising direction for future research.
>
> Thank you for suggesting AIGC data for evaluation. We conducted experiments using the T2VQA database [1] and would like to provide context before presenting our results:
> The T2VQA database has the following characteristics:
> - Each video is associated with a single Mean Opinion Score (MOS) aggregating ratings for text-video alignment and video fidelity, obtained through a 0-100 slider (Section 3.2 of [1]).
> - Human-evaluated scores exhibit significant variance, clustering around 50 with a standard deviation of 16.6 (Figure 3 and Section 3.2 of [1]).
> - The database contains generations from 10 different AI models. Many models yield similar MOS scores, resulting in an ambiguous ranking of the models if used directly (Figure 3 of [1]).
> - T2VQA-DB lacks a "ground-truth" video set. To address this, we identified corresponding videos from WebVid10M using the same prompts and utilized them as our ground-truth distribution for evaluation.
>
> We selected generations from three models (#2, 7, and 9) that exhibit a clear gap in video quality, with average rankings of 10 (worst), 5, and 1 (best), respectively, based on MOS scores. Both FVD and JEDi scores ranked these models consistently, with model #2 as the worst and model #9 as the best. Our evaluation results are
> |                                 | Model #2    | Model #7    | Model #9    |
> |---------------------------------|-------------|-------------|-------------|
> | Ranks according to MOS          | Worst          | Median           | Best           |
> | JEDi (#. conv samples) | 4.427 (100) | 2.164 (200) | 0.722 (300) |
> | FVD (#. conv samples)  | 884.1 (600) | 445.6 (800) | 279.9 (900) |
>
>  Importantly, JEDi achieved convergence with significantly fewer samples, compared to FVD: the number of samples to converge for JEDi are 100, 200 and 300; the number of samples to converge for FVD are 600, 800 and 900.
>
> We initially planned to evaluate JEDi and FVD on the LGVQ and GAIA datasets. However, the LGVQ dataset is not yet available, and GAIA lacks a ground-truth set, with its scores also reflecting action-tag alignment. We are currently exploring alternative AIGC datasets that offer decoupled visual quality scores from human raters alongside a ground-truth set. We will keep you updated on our progress.
>
> [1] Kou, T., Liu, X., Zhang, Z., Li, C., Wu, H., Min, X., Zhai, G., & Liu, N. (2024). Subjective-Aligned Dataset and Metric for Text-to-Video Quality Assessment. ArXiv, abs/2403.11956.

---

> ### Author Response · Authors · 2024-11-20
> **Follow-up**
>
> Dear Reviewer,
>
> Thank you for your valuable feedback. We have revised our manuscript accordingly:
> 1) Changing the paper title in OpenReview at this stage appears to be unfeasible; however, we have updated our title in the PDF to *Beyond FVD: Enhanced Metric for Evaluating Video Generation Distribution Quality*.
> 2) We have discovered a new dataset [1] containing additional AIGC samples. Specifically, TIP-I2V is comprised of samples generated by current state-of-the-art video generation models. Our analysis of these samples in the I3D and V-JEPA feature spaces reveals significant insights:
>     - Our findings are based on the principle that features extracted from video sets A and B should exhibit the same distribution if they originate from the same source.  This is because a consistent extractor should not introduce artificial distinctions between sets that inherently belong to the same distribution.
>     - On the other hand, if video sets A and B come from different distributions, a robust feature extractor designed for evaluating distributional differences would highlight these distinctions. A good extractor would encode the unique characteristics of each set, creating clearly separable distributions in the feature space. This separation is critical for understanding the underlying differences and verifying that the extractor is effectively capturing meaningful distributional variations rather than noise or irrelevant features.
>     - Our results in *Appendix G* demonstrate the superiority of V-JEPA feature space in this regard: we show that JEPA-extracted features reveal clear distributional distinctions between 3 AIGC video sets and 1 real video dataset, whereas I3D features do not exhibit such distinct clustering.
> 3) In the newly revised pdf, we have added a *new section (5.5)* focused on AIGC studies. This section presents the T2VQA-DB [2] results and discusses our findings on TIP-I2V [1]. A more detailed report on TIP-I2V is available in *Appendix G*.
> 4) We added another study on the effect of long-term video on our metric in *Section I, Pg 34*.
> 5) We have added a discussion of the implications of our findings for the single-video quality evaluation metrics in *Section J.2 (Pg 34-35)*.
>
> *[1] Wang, W., & Yang, Y. (2024). TIP-I2V: A Million-Scale Real Text and Image Prompt Dataset for Image-to-Video Generation.*
>
> *[2] Kou, T., Liu, X., Zhang, Z., Li, C., Wu, H., Min, X., Zhai, G., & Liu, N. (2024). Subjective-Aligned Dataset and Metric for Text-to-Video Quality Assessment. ArXiv, abs/2403.11956.*
>
> We hope these responses have addressed all of your concerns. We are also happy to discuss any remaining issues that may still be unclear. If we have clarified any confusion, we kindly ask that you consider revising your scores for our work.

---

### Official Review · Reviewer_cX2Z · 2024-11-04

**Soundness:** 2
**Presentation:** 2
**Contribution:** 2
**Rating:** 3
**Confidence:** 4

**Summary:**

This paper addresses the issues present in currently popular evaluation metrics in the field of video generation by proposing a new evaluation metric. It points out the assumptions underlying existing evaluation methods and the problems associated with these assumptions. The newly proposed metric shows better performance in terms of cosine similarity with human evaluations and demonstrates significant improvement in the number of samples needed to converge.

**Strengths:**

The paper comprehensively considers the strengths and limitations of existing metrics and systematically reviews the assumptions underlying these metrics, along with the problems associated with these assumptions. It delves into the accuracy of The Fréchet distance (FD) and its consequences when these assumptions do not hold. Building on this foundation, the paper addresses some of the issues present in current evaluation strategies. The new evaluation strategy proposed shows a significant improvement in consistency with Cosine Similarity with Human Evaluation.

**Weaknesses:**

The paper emphasizes that JEDi, as a video generation evaluation metric, has an advantage in the number of samples required for convergence, achieving faster convergence. However, the number of samples required for convergence should not be considered the primary concern of an evaluation strategy, as not all datasets face the issue of insufficient samples affecting precision; there are still many datasets that meet the necessary conditions. While JEDi performs well in consistency with human evaluations, further experimental validation is needed to assess its performance in other aspects as an evaluation metric.
Although the author said in the experiment that the resolution can be arbitrary, since this work relies on the representation capability of SreamDiffusion[3] based on SD-Turbo or LCM, many "high-resolution" images are not included in the training set of SD-Turbo or LCM. Can Promptus effectively transmit videos with various resolutions and high resolutions that it has not seen during SD training?

**Questions:**

Please refer to the Weakness section.
Additionally, the paper provides limited explanation of the proposed evaluation metric, JEDi. It might be beneficial to reduce the amount of reiteration about other evaluation metrics and instead increase the logical exposition of JEDi to provide a more comprehensive understanding of its strengths and limitations.

---

> ### Author Response · Authors · 2024-11-15
>
> Dear Reviewer,
>
> Thank you for your feedback and review.
>
> ```However, the number of samples required for convergence should not be considered the primary concern of an evaluation strategy, as not all datasets face the issue of insufficient samples affecting precision```
>
> In response to concerns about convergence issues, we clarify that sample deficiency is multifaceted: the sample convergence problem is not solely due to the number of available samples in different datasets; it also arises from the extremely slow generation of high-quality videos using modern generative video models (Section 4.2, lines 334-339). Most existing studies, for example [1], evaluate performance on just 256 videos on KTH, Cityscapes, and UCF-101 datasets. We have conducted UCF-101 analysis in Figure 3, and it needs more than 4000 videos to converge. Thus, 256 samples is extremely far from convergence. This limited sampling significantly hampers the metric’s ability to achieve convergence.
>
> In addition, as AIGC’s quality improves, their distribution increasingly aligns with the ground-truth distribution. Consequently, the margin of error in evaluating the distances between the two distributions decreases. This reduction necessitates both a larger number of samples and the use of a metric with higher sample efficiency to ensure reliable evaluation.
>
> These points demonstrate that our metric’s superior sample efficiency is a significant contribution.
>
> ```further experimental validation is needed to assess its performance in other aspects as an evaluation metric```
>
> Our paper presents an extensive comprehensive analysis, with some experiments included in the Appendix due to space constraints. To our knowledge, few existing works match this level of thoroughness. Our experimental scope encompasses:
> 1)  Non-Gaussianity assessment of Fréchet Distance across 5 architectures and 11 datasets (Figure 2).
> 2) Convergence evaluations for 5 metrics (with/without autoencoding) on 2-5 architectures and 11 datasets (Figures 14-15-16).
> 3) Noise sensitivity analysis for 5 metrics, 5 architectures, 2 datasets, and 10 visual variations (Figure 18).
> 4) Human preference comparisons across 50 combinations of 5 metrics, 5 architectures, and autoencoding configurations on 2 datasets (Table 4).
> 5) Temporal metric evaluations during video model fine-tuning for 4 metric-architecture pairs on 2 datasets (Figures 4-5).
>
> Building on our already extensive experimental evaluation, we will further supplement this rebuttal with additional analyses on:
> - T2VQA-DB, exploring the applicability of our findings to this benchmark dataset
> - Long-term video sequences, examining the temporal dynamics and robustness of our approach
> These supplementary analyses will provide even deeper insights into the effectiveness and the agreement to human subjects of our method.
>
> ```Although the author said in the experiment that the resolution can be arbitrary, since this work relies on the representation capability of SreamDiffusion[3] based on SD-Turbo or LCM, many "high-resolution" images are not included in the training set of SD-Turbo or LCM. Can Promptus effectively transmit videos with various resolutions and high resolutions that it has not seen during SD training?```
>
> We evaluate videos at the resolution required by the backbone network architectures, so we do not have control over the resolution of the videos. We are not familiar with the works you mentioned, nor how they relate to our metric. Could you please clarify your question?
>
> ```the paper provides limited explanation of the proposed evaluation metric, JEDi. It might be beneficial to reduce the amount of reiteration about other evaluation metrics and instead increase the logical exposition of JEDi to provide a more comprehensive understanding of its strengths and limitations```
>
> JEDi is the result of our extensive analysis of various statistical distribution metrics and backbone architectures across multiple datasets. This analysis is fundamental to justifying the development of JEDi. FVD, as a video distribution quality metric, uses Fréchet distance to measure the distance between video distributions in I3D space. While FVD and other related studies are valuable, it is important to acknowledge the limitations of these metrics and the feature space they operate in, and to improve upon them. We prefer to be transparent by:
> 1) showing all the methods we compared,
> 2) highlighting that the best combination was V-JEPASSv2+MMDPOLY (JEDi), and
> 3) proposing this combination as our metric.
>
> Our extensive experiments (see above) already compare JEDi against a broad range of other approaches, which underscores both its strengths and limitations.
>
> *[1] Voleti, V.S., Jolicoeur-Martineau, A., & Pal, C.J. (2022). MCVD: Masked Conditional Video Diffusion for Prediction, Generation, and Interpolation. ArXiv, abs/2205.09853.*

---

> ### Author Response · Authors · 2024-11-26
> **Follow-up**
>
> Dear Reviewer,
>
> As the manuscript revision period comes to a close, we have not yet received further feedback from you. We would greatly appreciate the opportunity to engage in more discussions with you during the extended discussion period.
>
> During the manuscript revision period, we have **added three new experiments**: two studies on AIGC datasets and one focusing on long-term video generation. Additionally, we have expanded our discussion to include details on the demographics of our human study and outlined potential avenues for future research. **A summary of our rebuttal updates is available in the global post.**
>
> For the discussion period, we would like to seek your feedback on the following points:
> - With the addition of three new experiments alongside the five existing ones, do you believe our work still lacks sufficient experimental validation?
> - **Could you clarify what you meant by your comment regarding `“high resolution”`, `“SreamDiffusion”`, `“SD-Turbo”`, `“LCM”`, or `“Promptus”`?**
> - Do you still feel our discussion on the strengths and limitations of our metric is sufficient now?
>
> We look forward to your insights and appreciate your time and effort in providing feedback.

---

> > ### Author Response · Authors · 2024-12-02
> > **Follow-up II**
> >
> > Dear Reviewer,
> >
> > **As the deadline for the author-reviewer discussion phase approaches, we are increasingly concerned as we have not yet received a response from you during the rebuttal period. In our previous two replies, we highlighted that certain aspects of your feedback seem unrelated to our work and requested clarification, but we have not yet heard back.**
> >
> > For your other comments, we hope that our previous replies have sufficiently addressed them. Additionally, we have thoroughly addressed the feedback from other reviewers and provided a summary in the global post. We kindly ask that you carefully review this when conducting your final evaluation of our work.
> >
> > Best regards

---

### Comment · Area_Chair_EeeL · 2024-11-25

Hi Reviewers,

We are approaching the deadline for author-reviewer discussion phase. Authors has already provided their rebuttal. In case you haven't checked them, please look at them ASAP. Thanks a million for your help!

---

### Author Response · Authors · 2024-11-26
**Rebuttal Summary**

Dear Reviewers,

We would like to extend our sincerest gratitude for your collaborative efforts in refining our work. Your constructive feedback has been invaluable, and we are confident that it has significantly enhanced the quality of our work.

**Below is a summary of the modifications made during the rebuttal period:**

### New experiments
1) STUDY ON ARTIFICIAL INTELLIGENCE GENERATED CONTENT: T2VQA-DB
    - This study utilizes the T2VQA database, comprising 10,000 AI-generated videos produced by 10 distinct Text-to-Video (T2V) AI models. Each video is associated with a human-rated Mean Opinion Score (MOS) that evaluates text-video alignment and fidelity.
We conducted an assessment of three T2V models from the T2VQA-DB dataset, representing a range of video qualities. Our analysis revealed that both FVD and JEDi scores consistently ranked these models in accordance with their MOS scores. Notably, JEDi demonstrated a significantly better sample efficiency rate.
    - *Section 5.5, Pg 9-10*
    - Suggested by Reviewer `goxo`, `NmJ7`, `gxbR`
2) STUDY ON ARTIFICIAL INTELLIGENCE GENERATED CONTENT: TIP-I2V
    - This study investigates the outputs of three state-of-the-art video generation models and a real video dataset’s samples within the FVD and JEDi embedding spaces. Our analysis reveals distinct distributional patterns in the features extracted by V-JEPA (used in JEDi), whereas features extracted by I3D (used in FVD) do not exhibit such clear distinctions. These findings suggest that V-JEPA's feature space is more suitable for video evaluation
    - *Appendix H, Pg 33*
    - Suggested by Reviewer `goxo`
3) LONG-TERM VIDEO STUDY
    - We investigate the impact of clip length (16, 32, 64, 128 frames) on metric performance. Our findings show that the JEDi metric is sensitive only to the level of distortion, providing consistent evaluations across different clip lengths under the same distortion. In contrast, FVD is influenced by clip length, with distributional distances increasing as the clip length grows, even when the distortion level remains unchanged.
    - *Appendix I, Pg 34*
    - Suggested by Reviewer `uZfY`

### New Discussions
1) Study on Artificial Intelligence Generated Content: *Section 5.5, Pg 9-10* [`goxo`, `NmJ7`, `gxbR`]
2) Computation Configuration: *Appendix B, Pg 17-18*
3) Human Study Demographics: *Appendix F.3, Pg 29* [`uZfY`, `NmJ7`, `goxo`]
4) Survey on Human Studies in Previous Literature: *Appendix F.4, Pg 30-32* [`NmJ7`, `goxo`]
5) Avenues for Future Researches (Implication for Single Video Assessment Metrics): *Appendix J.2, Pg 34-35* [`goxo`, `gxbR`]

### Revised Figures
We have fixed *Figure 2, 3, 7, 8, 9, 10, 11, 12, 13, 14, 15, 16, 17, 18, 19* suggested by Reviewer `NmJ7`

Thank you again for your insightful feedback and helpful suggestions. Your input has greatly improved our document, and we appreciate this opportunity to work with you on our paper's rebuttal.

Authors

---

### Meta-Review · Area_Chair_EeeL · 2024-12-20

**Metareview:**

This paper finds limitations of FVD for evaluating video generation distribution quality and proposes JEDi based on features derived from a Joint embedding predictive architecture, measured using maximum mean discrepancy with polynomial kernel. Experiments on several datasets show the proposed metrics is better than FVD.

This paper got 5 reviews with mixed scores, i.e., 2 accept, 2 marginally above the acceptance threshold and 1 reject.

The strength of this paper are: 1) proposed JEDi eliminates the need for parametric assumptions about the underlying video distribution; 2) proposed methods significantly reduces the number of samples needed; 3) better aligned with human evaluations; 4) limitations of FVD are systematically identified; 5) experimental results are solid.

Weaknesses are: 1) proposed JEDi is a distribution-based evaluation metric, can not work for evaluating a single generated video; 2) more evaluation on existing large-scale AIGC video quality datasets should be conducted; 3) more comparisons with the single AIGC video quality assessment metrics are suggested to be included; 4) paper structure is not good; 5) need more experimental results; 6) missing analysis on long-term temporal distortions;

In the rebuttal, Reviewer goxo gave score 8 and didn't reply. Reviewer NmJ7 thought the authors addressed their concerns quite well and changed rating to 8. Reviewer gxbR also said their concerns are addressed and changed score to 8. Reviewer uZfY maintained positive rating 6. Reviewer cX2Z’s gave score 3 and didn't reply. AC checked reviewer cX2Z's comments and author's rebuttal and did find some parts are irrelevant. The rest are addressed by authors. Given these, AC decided to accept this paper.

**Additional Comments On Reviewer Discussion:**

In the rebuttal, Reviewer goxo gave score 8 and didn't reply. Reviewer NmJ7 thought the authors addressed their concerns quite well and changed rating to 8. Reviewer gxbR also said their concerns are addressed and changed score to 8. Reviewer uZfY maintained positive rating 6. Reviewer cX2Z’s gave score 3 and didn't reply. AC checked reviewer cX2Z's comments and author's rebuttal and did find some parts are irrelevant. The rest are addressed by authors. Given these, AC decided to accept this paper.

---

### Decision · Program_Chairs · 2025-01-22

Accept (Poster)